# Novel Receptor Tyrosine Kinase Pathway Inhibitors for Targeted Radionuclide Therapy of Glioblastoma

**DOI:** 10.3390/ph14070626

**Published:** 2021-06-29

**Authors:** Julie Bolcaen, Shankari Nair, Cathryn H. S. Driver, Tebatso M. G. Boshomane, Thomas Ebenhan, Charlot Vandevoorde

**Affiliations:** 1Radiobiology, Radiation Biophysics Division, Nuclear Medicine Department, iThemba LABS, Cape Town 7131, South Africa; snair@tlabs.ac.za; 2Radiochemistry, South African Nuclear Energy Corporation, Pelindaba, Brits 0240, South Africa; cathryn.driver@necsa.co.za; 3Pre-Clinical Imaging Facility, Nuclear Medicine Research Infrastructure, Pelindaba, Brits 0242, South Africa; thomas.ebenhan@up.ac.za; 4Department of Nuclear Medicine, University of Pretoria Steve Biko Academic Hospital, Pretoria 0001, South Africa; gboshomane@gmail.com; 5Preclinical Drug Development Platform, Department of Science and Technology, North West University, Potchefstroom 2520, South Africa

**Keywords:** targeted radionuclide therapy, glioblastoma, radiochemistry, theranostics, molecular imaging, tyrosine kinases, radiopharmaceuticals

## Abstract

Glioblastoma (GB) remains the most fatal brain tumor characterized by a high infiltration rate and treatment resistance. Overexpression and/or mutation of receptor tyrosine kinases is common in GB, which subsequently leads to the activation of many downstream pathways that have a critical impact on tumor progression and therapy resistance. Therefore, receptor tyrosine kinase inhibitors (RTKIs) have been investigated to improve the dismal prognosis of GB in an effort to evolve into a personalized targeted therapy strategy with a better treatment outcome. Numerous RTKIs have been approved in the clinic and several radiopharmaceuticals are part of (pre)clinical trials as a non-invasive method to identify patients who could benefit from RTKI. The latter opens up the scope for theranostic applications. In this review, the present status of RTKIs for the treatment, nuclear imaging and targeted radionuclide therapy of GB is presented. The focus will be on seven tyrosine kinase receptors, based on their central role in GB: EGFR, VEGFR, MET, PDGFR, FGFR, Eph receptor and IGF1R. Finally, by way of analyzing structural and physiological characteristics of the TKIs with promising clinical trial results, four small molecule RTKIs were selected based on their potential to become new therapeutic GB radiopharmaceuticals.

## 1. Introduction

Gliomas represent 80% of all primary brain tumors and form a heterogeneous group of tumors of the central nervous system (CNS). Glioblastoma (GB, WHO grade 4) is the most frequently occurring malignant CNS tumor with a global incidence of 0.59–3.69 per 100,000 [1,2]. Nowadays, the WHO classification no longer relies solely on histological criteria but has also implemented additional molecular biomarkers for classification, including isocitrate dehydrogenase (IDH) mutation, 1p/19q co-deletion, H3-K27M mutation and O6-methylguanine DNA methyltransferase (MGMT) promoter methylation [3]. The current standard of care consists of maximum safe surgical resection followed by external beam radiation therapy (EBRT) plus concomitant and adjuvant temozolomide (TMZ). However, GB has a poor prognosis with a median survival of approximately 14 months and less than 10% of patients living longer than 5 years from diagnosis. At recurrence, there is no consensus on the standard of care as no therapeutic options thus far could demonstrate a substantial survival benefit [4,5,6]. IDH mutation and 1p/19q co-deletion are the most prognostically favorable molecular markers, but regrettably, most GBs are IDH wild-type and the benefit from TMZ is most prominent in patients with MGMT promoter-methylated GB [3,7,8].

Treatment challenges posed by malignant gliomas include molecular and cellular heterogeneity, innate and acquired therapy resistance and the obstacle of effective drug delivery posed by the blood–brain barrier (BBB) [9,10,11]. An improved understanding of the underlying disease pathology and the causes for these treatment challenges might aid the development of new GB therapy strategies. One particular strategy is the development of theranostic agents that combine diagnostic molecular imaging with therapy using the same agent. The theranostic agent thereby investigates the presence of a certain target on the tumor cells of the patient while the therapeutic version of the agent (commonly a radioactive derivative) binds to the same target and induces tumor cell death by emitting radiation, while sparing healthy normal tissues. The latter approach is called targeted radionuclide therapy (TRT) [12,13]. Our group recently published a perspective on the requirements for a successful TRT agent for GB treatment and the existing TRT strategies for GB were recently reviewed by others [14,15,16,17].

In this review, the option to use the tyrosine kinase (TK) pathway as a target for GB radiopharmaceutical development, and specifically for TRT, will be explored. RTKs are a family of cell surface receptors that contain a ligand-binding region in the extracellular domain, a single trans-membrane helix, and a cytoplasmic region that contains the protein TK domain plus additional carboxy (C-) terminal and juxtamembrane regulatory regions [18]. Their extracellular ligand-binding domain acts as a receptor for growth factors, hormones, cytokines, neurotrophic factors and other extracellular signaling molecules [1]. Upon binding, RTKs mediate cell-to-cell communication, cell growth, motility, differentiation, cell cycle control and metabolism [19]. Mutations in RTKs and aberrant activation of their intracellular signaling pathways have been linked to malignant transformation and have driven the development of a new generation of drugs that block or attenuate RTK activity [18,19]. There are 62 Food and Drug Administration (FDA)-approved therapeutic agents that target about two dozen different protein kinases, most of which are classified as receptor tyrosine kinase inhibitors (RTKI) [20]. RTKIs consist of mainly two categories, monoclonal antibody-based drugs (mAbs) that bind to the extracellular domain of the receptor and small molecule inhibitors (SMIs) acting intracellularly, both of which result in blocking the downstream signal transduction cascade [21,22,23]. Aberrant RTK activation is a well-studied therapeutic target in GB, but despite the tremendous effort in TKI development, results are variable and complicated by treatment resistance [24,25].

Personalized medicine strategies, such as the use of theranostic agents, could assist in improving the effectiveness of TKI treatment strategies by improving patient selection, providing a more complete pathway inhibition, improving tumor radiosensitization and the prediction of the treatment response [23,26,27]. Therefore, this review will give an overview of the current status of TKIs and TKI radiopharmaceuticals for GB applications. By way of analyzing structural and physiological characteristics of the TKIs with promising clinical trial results, this review will suggest suitable candidates that are most promising to become radiopharmaceuticals for TRT of GB.

## 2. Nuclear Molecular Imaging and TRT Using TKIs

Due to limited target expression/engagement, mutation status and unfavorable pharmacokinetic (PK) parameters of TKIs, only a minority of patients will ultimately benefit from TK targeted therapy [28]. Moreover, the initial response rates are hampered by drug resistance that occurs over the course of treatment. As such, the identification of treatment-responsive patients constitutes one of the key challenges associated with the clinical use of TKIs. Traditionally, biopsy and immunohistochemistry are performed to determine the RTK status of cancer tissues and to guide subsequent treatment plans. However, spatial expression levels of RTKs can vary over time and amongst lesions [27]. Nuclear imaging of TK expression using Single-Photon Emission Computed Tomography (SPECT) and Positron Emission Tomography (PET) could therefore play an important role for subsequent TKI therapy [27,28]. Hence, non-invasive PET/SPECT imaging may be utilized to measure TK activity distribution in vivo, which in turn can be used to determine patient-specific treatment plans with calculated dosages to target and non-target organs (image-based dosimetry) [29]. Additionally, it can add a better understanding of the spatial RTK patterning in the very heterogeneous GB tumor microenvironment and allow for the longitudinal behavior of drugs (FDA-approved) to be studied in vivo [23,27,28]. Commonly used PET and SPECT radionuclides are listed below and can be divided into either organic (Table 1) or metallic (Table 2) radionuclides. The application of organic radionuclides for diagnostic purposes has been centered around the use of fluorine-18 with its favorable half-life and positron emission, while therapy has successfully been completed using iodine-131 with its suitable beta emissions. However, the widespread application and development of new radiopharmaceuticals using these organic radionuclides has been limited by their availability and the need for long and complicated radiosynthesis to incorporate these nuclides covalently into the molecules of choice [30]. The list of radiometals being used is extensive across both diagnostic and therapeutic areas. Radiometals are bound to the desired compounds through coordination with chelating donor-ligands to form inert metal-chelate complexes, most often in one-pot reactions. This process is therefore much simpler than the complex syntheses required for organic radiolabeling and is resulting in radiometals being favored for use in diagnostic and therapeutic applications and increasingly researched for new product developments. In 2020, the FDA had approved 51 radiopharmaceuticals for the market of which only 19 (37%) were non-metallic isotopes [31].

FDA-approved RTKIs generally fall into the category of either mAbs/proteins (>1 kDa) or small molecules (SM) (<1 kDa). Strategies for radiolabeling these inhibitors are dependent on the desired usage (imaging and/or therapy) and therefore the combination with adequate radionuclide(s), and particular characteristics of the inhibitor itself. The radionuclide properties to be considered include the type of radiation and emission properties (including the linear energy transfer), the radionuclide half-life, chemical purity, specific activity and most importantly, availability and production cost of the radionuclide [17,34,35]. Characteristics of the inhibitor to be factored in include the biological half-life and blood clearance of the targeting ligand, the affinity of the inhibitor for its target, the available positions for radiolabeling and the proposed drug metabolism [28]. Consequently, the radionuclide properties should be matched with the TKI properties in order to result in a functional and effective radiopharmaceutical.

The size of the mAbs and challenging radiosynthesis parameters generally preclude the use of direct labeling with common non-metallic radionuclides (such as carbon-11 and fluorine-18); however, indirect labeling with iodine-123/-125/-131 has been accomplished [36,37]. Thus, radiolabeling of mAb/protein TKIs is most often accomplished by chelation of a radiometal isotope since mAbs can be easily modified to contain a chelating agent for facile metal complexation [38]. Furthermore, these types of complexation often occur rapidly at mild temperatures in biocompatible solutions that are ideal for protecting the integrity of the mAb [30]. There are a number of suitable radiometal isotopes with longer half-lives to match the extended biological half-lives of antibodies—the most favorable kind (for imaging purposes in this regard) is zirconium-89. Conventional strategies for linking chelating agents covalently to mAbs include reaction of electrophilic isothiocyanate and activated carboxyl groups (*N*-hydroxysuccinimide esters and anhydrides) with nucleophilic amino groups of accessible lysines, and the Michael-type addition reaction of maleimide groups with any available thiols in the mAb. A number of new site-specific conjugation techniques to mAbs are also being developed [38]. Chelating agents for radiometal isotopes and their attachment to biomolecules have been extensively reviewed [39,40,41,42]. The selected chelating agent needs to bind the selected radionuclide with high thermodynamic stability and kinetic inertness to ensure high stability of the complex in vivo [30]. In vivo complex stability is very crucial to prevent innate transchelation of the radiometal to a number of endogenous competitive metal-binding proteins. The advantage of mAb TKI’s (high affinity for target) is offset by the disadvantage of long metabolic cycles and their molecular size and hydrophilicity which limits their BBB penetration. This issue may be overcome by smaller antibody fragments or affibody molecules that mimic antibody-binding properties [17,43]. For radiolabeling of mAb fragments, the short-lived positron emitters, such as gallium-68 (t½ 1.13 h), copper-64 (t½ 12.7 h), yttrium-86 (t½ 14.7 h) and bromine-76 (t½ 16.2 h) are available to facilitate in vivo imaging. Various strategies on the efficient radiolabeling of mAb have been improved and are continually reviewed [44,45,46,47].

The majority of the FDA-approved TKIs are SM probes since they are generally considered safe, have a favorable PK and show high affinity and specificity, but they are inadequate for delayed imaging after injection due to their fast clearance [48]. In contrast to the radiolabeling of mAbs, radiolabeling of TK SMIs is mostly through covalent incorporation of non-metallic radionuclides (carbon-11, fluorine-18) and is therefore more challenging and requires a drug-specific labeling strategy. The radiosynthesis employed needs to achieve the product in a short time, with sufficient yields and with the highest possible specific activity. A number of the TKIs carry readily accessible methoxy (-O-CH3) or amine (-N-CH3) moieties for radiolabeling with carbon-11 through conventional radiomethylation methods. Others carry fluorine atoms in activated ortho- or para-aryl positions or non-activated fluoro-aryl moieties that might allow for conversion to fluorine-18; other less suitable radiolabeling positions would include asymmetrical ureas and tolyl groups [28]. The advantage of carbon-11 and fluorine-18 for PET imaging purposes is that the radiolabeled TKI structure matches the original TKI and will therefore have exactly the same PK properties, and the short half-life of the radionuclide is suited to the fast clearance rates of the TKI.

TKIs, mAb and SMI have previously been radiolabeled for diagnostic purposes, some also for use in TRT, with progress in the field of epidermal growth factor receptor (EGFR) radio-immunotherapy (RIT) [23,27,28,49,50,51,52]. For instance [^188^Re]Re-nimotuzumab and [^125^I]iodo-mAb 425 are in a clinical stage for GB and [^177^Lu]Lu]-/[^211^At]At-L8A4 (EGFRIII) mAb and [^177^Lu]Lu-cetuximab are in a preclinical stage for GB [37,53,54,55,56,57,58]. In addition, RTK signaling is affected by ionizing radiation and hence the combination of TKI and RT (external beam radiation therapy or TRT) could have synergistic effects [59].

## 3. Receptor Tyrosine Kinase Inhibitors (RTKIs) for GB Therapy

In this review, the focus will be on seven RTKs, based on their central role in GB: the EGFR, the vascular endothelial growth factor receptor (VEGFR), the mesenchymal-epithelial transition factor (MET) receptor, the platelet-derived growth factor receptor (PDGFR), the fibroblast growth factor receptor (FGFR), ephrin receptor (Eph-receptor) and the insulin-like growth factor 1 receptor (IGF1R) [1,60,61,62,63,64]. Aberrations and gene expression of EGFR, neurofibromatosis 1 (NF1) and PDGFRA/IDH1 each define classical, mesenchymal and proneural GB, respectively [65]. Upon activation by ligands, these RTKs signal through two major downstream pathways: rat sarcoma (Ras)/mitogen-activated protein kinase (MAPK)/extracellular signal-regulated kinase (ERK) and Ras/phosphoinositide 3-kinase (PI3K)/protein kinase B (Akt) (Figure 1) [1]. These pathways are involved in the regulation of cell proliferation, survival, differentiation and angiogenesis. The Cancer Genome Atlas (TCGA) confirmed that genetic alterations in the RTK/Ras/PI3K pathway are present in up to 88% of GB tumors [9,66]. The PI3K pathway is also influenced by other aberrant signaling cascades, including loss of function of the phosphatase and tensin homologue (PTEN) protein, amplification and/or mutation of the EGFR, which occurs in 40% and 50% of GB cases, respectively [66].

Numerous drugs that inhibit RTKs and their signaling cascades downstream are currently in phase I and II clinical trials, either in combination with other FDA-approved chemotherapeutic agents such as TMZ, or as emerging, monotherapeutic agents [67,68]. Figure 1 gives an overview of RTKIs that reached the clinic for the treatment of GB.

### 3.1. Epidermal Growth Factor Receptor (EGFR)

#### 3.1.1. Current Status of EGFR and EGFRIII Targeted Therapy in GB

EGFR genetic alterations, including mutations, rearrangements, alternative splicing and focal amplifications are the dominant RTK lesions in GB. They occur in 57% of tumors and, overall, are the most common oncogene alteration in GB [69]. The most frequent genetic aberration associated with malignant glioma is an amplification of the EGFR, also referred to as ERBB1 or human epidermal receptor 1 (HER1), and the expression of the EGFR variant III (EGFRvIII). Updated insights on EGFR signaling pathways in glioma have recently been reviewed [70]. Due to the important role of the EGFR pathway in glioma, the interest in therapeutically targeting EGFR increased rapidly over the past few decades. Numerous clinical trials targeting EGFR have been completed in GB patients, but unfortunately with disappointing results—see Table 3 [69,70,71,72,73,74]. There are two predominant classes of EGFR inhibitors: (1) SMIs that target the receptor catalytic domain of EGFR, such as gefitinib and erlotinib and (2) mAbs that target the extracellular domain of EGFR, such as cetuximab and nimotuzumab [75].

The first-generation EGFR inhibitors that sterically block the ATP/substrate-binding pocket of EGFR include gefitinib, erlotinib and lapatinib, but all three did not result in improved overall survival (OS) for primary and recurrent GB as a monotherapy or in combination therapies [76,77,78]. The second-generation inhibitors were designed to irreversibly bind to the TK domain of EGFR and other ERBB family members and include afatinib and dacomitinib, both FDA-approved. The combination of afatinib and TMZ showed anti-GB effects preclinically but limited single-agent activity was observed in recurrent GB patients. In the most recent retrospective trial on dacomitinib, a small subset of patients (14.3%) showed clinical benefits [79,80,81]. The third-generation EGFR inhibitors include AZD9291 (osimertinib) and AEE788 (everolimus). Preclinically, AZD9291 proved to have better activity and selectivity for GB than the previous inhibitors, thereby overcoming primary resistance by continuously blocking ERK signaling [82,83]. In a case report, AZD9291 demonstrated clinical activity and a phase II study in recurrent GB patients is currently in the recruitment phase (NCT03732352) [67,84]. *AEE788* also inhibits VEGFR, next to EGFR, but was associated with unacceptable toxicity and minimal activity, despite promising results in animal models [85,86,87].

**Table 3 pharmaceuticals-14-00626-t003:** Clinical trials in GB targeting the epidermal growth factor receptor.

Compound	Type	Clinical Trials: Phase, Overall Conclusion (+) or (−), (Combined Therapy)	Reference
Gefinitib (ZD1839)	SM	II (−) (after RT/chemo)	[88]
		II (−) (RT)	[76]
		I/II (−) (RT)	[89]
		II (+/−) (cediranib)	[90]
Erlotinib (Tarceva, OSI-774)	SM	I (+) (RT)	[91]
		II (−)	[92]
		II (−) (RT + TMZ)	[93]
		I/II (−) (single)	[94]
		II (−) (single)	[95]
		II (−) (TMZ/carmustine)	[78]
		II (−) (sirolimus)	[77]
		II (−) (RT/TMZ/bevacizumab)	[96]
		II (−) (carboplatin)	[97]
		Pilot (ongoing) (sunitinib, vandetanib)	NCT02239952 [67]
Lapatinib (GW572016)	SM	See Table 9	
Afatinib (Tovok, BIBW2992)	SM	I/II (−) (TMZ)	[79]
		I (ongoing)	NCT02423525 [67]
Dacomitinib (Vizimpro, PF299804)	SM	II (−) (single)	[98]
		II (retrospective, subset +)	[80]
Vandetanib (Caprelsa, ZD6474)	SM	See Table 9	
Tesevatinib (KD019/ XL647)>	SM	See Table 9	
Osimertinib (AZD9291)	SM	II (recruiting)	NCT03732352 [67]
Everolimus (AEE788)	SM	See Table 9	
Cetuximab (IMC-C225, Erbitux)	mAb	II (−)	[99]
		II (−) (bevacizumab, irinotecan)	[100]
		I/II (RT/TMZ)	[101]
		II (ongoing) (RT)	NCT02800486 [67]
		I/II (ongoing) (mannitol)	NCT02861898 [67]
Nimotuzumab (OSAG101)	mAb	II (+) (RT/chemo)	[102]
		I/II (+) (RT)	[103]
		I/II (+) (RT/chemo)	[104]
		I/II (+/−) (RT/TMZ)	[105]
		II/III (+) (RT)	[106,107]
		III (+/−) (RT/chemo)	[108]
Panitumumab (Vectibix, ABX–EGF)	mAb	II (−) (irinotecan)	NCT01017653 [67]
GC1118	mAb	II (ongoing)	NCT03618667 [67]
Depatuxizumab mafodotin (ABT-414/mAb 806)	Ab-drug	I (+/−) (single)	[109]
		I (+) (TMZ)	[110]
		I (+) (RT/TMZ)	[111]
		I (+) (TMZ)	[112]
		II (x) (TMZ/lomustine)	[113]
		II/III (ongoing) (RT/TMZ)	NCT02573324 [67]
ABT 595	Ab-drug	I (+)	[114]
Epitinib (HMPL-8)	SM	I (ongoing)	NCT03231501 [67]
Rindopepimut (CDX110)	Vaccine	II (+) (TMZ)	[73]
		III (−) (TMZ)	[115]
		II (+) (bevacizumab)	[116]
CART-EGFRvIII T	CARs	I (terminated)	[117]
		I Pilot (−)	[118]
Anti-CD3/EGFR Bispecific Antibody Armed T Cells (EGFR BATs)	bAb-T	I (RT/TMZ) (ongoing)	NCT03344250 [67]
T Cells (EGFR BATs)			
EGFR(V)-EDV-Dox	EDV	I (ongoing)	NCT02766699 [67]
AMG 596	BiTE	I (single/AMG 404) (ongoing)	NCT03296696 [67]
Sym004	Ab mix	II (completed, no results)	NCT02540161 [67]

Abbreviations: bAb (bispecific antibody), bispecific T cell engager (BiTE^®^) antibody construct, CARs (chimeric antigen receptors), doxorubicin (Dox), EnGeneIC delivery vehicle (EDV), T (T cells).

Cetuximab, panitumumab and nimotuzumab are FDA-approved anti-EGFR antibodies that bind to the L2 domain, preventing ligand binding and/or dimerization of EGFR [119]. While both cetuximab and panitumumab failed to demonstrate efficacy for recurrent GB, preclinical and multiple clinical trials with nimotuzumab in high-grade glioma patients gave promising results [102,103,107,108].

Other EGFRvIII-specific or preferential agents are in development, stimulated by the fact that the EGFRvIII variant, in contrast with wild-type EGFR, is not responsive to antibodies targeting the L2 domain because of the deletion mutation in the ligand-binding domain [120]. Two antibody drug conjugates have been evaluated for GB, including depatuxizumab mafodotin (ABT-414) and AMG 595. ABT-414 consists of mAb 806 and has shown encouraging but mixed results and grade 1/2 ocular toxicities occurred. Further studies and results are currently under way to evaluate its efficacy in EGFR-amplified, newly diagnosed and recurrent GB (NCT02573324, NCT02343406) [67,109,110,111,112]. AMG 595 was very effective in GB xenograft animal models and had favorable results during its phase I clinical study [114,121].

EGFRvIII-specific vaccines have also been evaluated as a way to activate the host immune system and provide durable responses in GB. However, positive phase II results of the peptide-vaccine rindopepimut could not be confirmed in a phase III randomized study (ACT IV) [73,115]. The randomized trial of rindopepimut combined with bevacizumab supports the potential for GB targeted immunotherapy, but the therapeutic benefit requires validation [116].

Clinical studies have so far failed to prove that EGFR is a reliable prognostic marker, despite recent confirmation that EGFRvIII sensitizes a fraction of GB patients to current standard of care treatment through the upregulation of DNA mismatch repair [122]. Unfavorable results are partly related to resistance mechanisms, categorized as resistant gene mutations, activation of alternative pathways, phenotypic transformation and resistance to apoptotic cell death [22]. In addition, unlike the kinase domain alterations seen in non-small-cell lung carcinoma (NSCLC), EGFR alterations in GB lie primarily in the extracellular domain. SMIs are difficult to develop for the extracellular domain, while mAbs are easier to design but they contend with the BBB [123]. Further studies with novel agents or combination strategies are warranted to re-evaluate the value of EGFR inhibition in molecularly selected GB populations [124]. Tools to identify subgroups of GB patients with true EGFR-dependency are urgently needed. A first step has been made in the retrospective study of Ronellenfitsch et al., where Akt and mTORC1 signaling was found to be a predictive biomarker for the EGFR antibody nimotuzumab in GB [125].

#### 3.1.2. EGFR Radiopharmaceuticals

EGFR molecular imaging is an optimal method for evaluating EGFR expression, which has been shown to be variable in GB, and for determining the EGFR mutation status in vivo [115,126]. Ideally, all future clinical studies should include pre-/post-treatment evaluation of EGFR expression and measurement of intra-tumoral drug concentration. In addition, known escape pathways (such as PI3K and MET) would ideally be assessed as well [123]. Advances in the development of EGFR-targeted molecular cancer imaging agents, including EGF ligands, mAbs, antibody fragments, affibodies and SMs were recently reviewed [48].

The 4-anilinoquinazoline scaffold is considered a privileged structure and serves as the core framework for several SMIs of EGFR [127]. This quinazoline scaffold is known as the hinge-binding motif as its hydrogen bonds to the hinge region of the ATP-binding pocket while the 4-anilino group with varying substituents is orientated in a way that it forms hydrophobic interactions with an adjacent back pocket [128]. Improvement of TK selectivity and development of new EGFR inhibitors was achieved through structure activity relations analysis of this 4-anilinoquinazoline scaffold with modifications to the substituents. Radiolabeling of these inhibitors therefore also occurs at sites attached to the quinazoline scaffold in order to maintain receptor binding integrity. To image EGFR expression-activity in GB, TKIs (SM and antibodies) have strategically been radiolabeled with [^11^C], [^18^F], [^64^Cu] or [^89^Zr] for PET and with [^131^I] or [^111^In] for SPECT. EGFR inhibitors were also labeled with therapeutic isotopes, such as [^125^I], [^211^At], [^177^Lu] and [^188^Re] (Figure 2 and Appendix A) [14,15,54,55,129]. Reversible inhibitors of EGFR TK labeled with isotopes of iodine or [^99m^Tc] were designed to be longer-lived radiopharmaceuticals. Although longer half-lives may give better distribution kinetics, the increase in lipophilicity and size of incorporating iodine or metal chelates into the quinazoline scaffold likely hindered the retention of these compounds into the binding pocket of EGFR TK [48,51].

[^11^C]C-PD153035, a potent and specific ATP-competitive TKI of the EGFR was capable of visualizing 6 of the human 8 GB tumors using PET and demonstrated favorable biodistribution and radiation dosimetry [130,131]. The concept of using [^111^In]In-EGFRvIII-CAR labeled T cells is undergoing clinical evaluation for newly-diagnosed GB and intracerebral EGFR-vIII-CAR T cells for recurrent GB; however, one of the trials was terminated (NCT02664363, NCT03283631) [67]. PET imaging studying [^11^C]C-erlotinib in glioma xenografts showed specific binding of the radiopharmaceutical for activating mutations of the kinase domain but no specific binding for activating mutations of the extracellular domain of the EGFR [132]. Important for GB is that the distribution of [^11^C]C-erlotinib was affected by two efflux transporters expressed on the BBB, ATP-binding cassette transporter G2 (ABCG2; also known as breast cancer resistance protein, BCRP) and P-glycoprotein (ABCB1), which are known to restrict successful drug delivery [28,133]. A brain distribution study using [^11^C]C-erlotinib-PET showed that co-infusion of erlotinib/tariquidar may potentially allow for complete ABCB1/ABCG2 inhibition, while simultaneously achieving brain-targeted EGFR inhibition [134]. A [^18^F]-radiolabeled erlotinib also showed uptake in nHepG2, HCC827, NSCLC and A431 tumor xenografts (Appendix A) [135,136,137].

Gefinitib (Iressa^®^, Astra Zeneca, Cambridge, UK) was radiolabeled with [^11^C] and [^18^F], but no increased uptake was seen after [^18^F]F-gefitinib injection into a GB animal model, although treatment with ABCB1/ABCG2 inhibitors led to enhanced brain penetration [136,138,139,140,141]. Other noteworthy groups of radiopharmaceuticals that derived from the 4-(anilino)quinazoline pharmacophore are [^18^F]F-ML01 [142], [^11^C]C-ML03 [143,144], [^11^C]C/[^18^F]F-ML04 [144,145] and [^18^F]F-PEG4-ML04 [146,147]. [^18^F]F-ML04 showed uptake in U87MG wild-type EGFR tumors which was at least in part, EGFR-associated. However, the distribution of the radiopharmaceutical was flow-limited, warranting possible modifications in chemical structure, as well as in the route of administration [145]. Pantaleo et al. failed to demonstrate the accumulation of different PEG-ylated anilinoquinazoline derivatives labeled with [^124^I], [^18^F] and [^11^C] in subcutaneous GB xenografts in mice [148]. [^131^I]IPQA, which binds specifically to activated EGFR kinase, showed rapid accumulation and progressive retention post washout in U87MG GB cells with a EGFRvIII mutant receptor [147]. IPQA was also radiolabeled with [^124^I] and [^18^F] with positive preclinical results by visualizing NSCLC and epidermoid carcinoma A431 in vivo [147,149,150].

Radiolabeled EGFR antibodies include mAb 425, nimotuzumab, mAb 806, cetuximab and mAb L8A4. Good results were obtained by applying [^125^I]iodo-mAb 425-RIT, either as monotherapy or in combination with TMZ, which was well-tolerated in patients and prolonged survival in a phase II clinical trial [55,129,151]. Single dose intracavitary [^188^Re]Re-nimotuzumab showed no improvement of median survival in a phase I clinical trial in 8 GB patients [54]. [^64^Cu]Cu-DOTA-cetuximab-PET and [^111^In]In-ABT-806- or [^111^In]In-cetuximab-SPECT showed uptake in intracranial GB models [152,153,154]. [^89^Zr]Zr-cetuximab reached the clinic (head and neck cancer and colorectal cancer) and cetuximab has also been labeled with [^125^I], [^88^Y] and [^177^Lu] to allow a theranostic approach; however, this was not applied for GB yet (Appendix A) [155,156,157,158]. The radiopharmaceutical [^64^Cu]Cu-/[^177^Lu]Lu-cetuximab seemed especially useful as a diagnostic tool for patient selection and as a potent RIT agent in EGFR-positive esophageal squamous-cell carcinoma [158]. Interestingly, a cetuximab Fab has been explored to overcome the unfavorable PK of the full-length mAb and was labeled with [^111^In] and [^64^Cu], but is not yet studied in GB (Appendix A) [159,160]. The EGFRIII mAb L8A4 was radiolabeled with [^125^I], [^177^Lu], [^211^At] and [^131^I], with possible therapeutic opportunities. Multiple preclinical studies in GB were performed [37,56,57,58,161,162,163]. Animals received [^125^I]iodo-BD-L8A4 by either convection enhanced delivery (CED) or direct intratumoral injection. Therapeutic efficacy was shown using either boronated mAb L8A4 alone or in combination with boronophenylalanine in the F98 GB model [163]. Different [^177^Lu]-labeled conjugates of L8A4 were compared to the characteristics of [^125^I]iodo-SGMIB-L8A4 in GB in vivo and in vitro [37,56,57].

Moreover, the single chain EGFR targeting antibody fragment [^125^I]iodo-SIPC-MR1(scFv) and radiolabeled affibodies ([^89^Zr]Zr-nimotuzumab, [^18^F]FBEM-Cys-ZEGFR:1907) showed tumor retention in GB in vivo [164,165]. [^89^Zr]Zr-panitumumab is a novel immuno-PET radiopharmaceutical (EGFR/HER1) and first clinical trials confirm that safety and dosimetry estimates were reasonable for clinical imaging but did not include GB patients [166,167,168,169,170,171].

Apart from EGFR mAbs, EGF-based ligand imaging probes are also designed. Some of these agents have been reviewed by Chen et al. [48]. EGF ligands were radiolabeled with [^131^I], [^68^Ga], [^18^F] and [^111^In], but data in GB are limited (Appendix A) [166,172,173]. Only [^111^In]In-benzyl-DTPA-hEGF resulted in positive in vitro results for GB [174]. An EGFRvIII-targeting peptide, H-Phe-Ala-Leu-Gly-Glu-Ala-NH2 (FALGEA), was also studied preclinically in GB; [^18^F]FBA-FALGEA accumulated preferentially in human GB xenografts [175]. Finally, the [^188^Re]-labeled DNA aptamer U2, targeting U87MG-EGFRvIII, dramatically inhibited tumor volume in a mouse model bearing U87MG GB [176].

Other EGFR immuno-PET and TKI-PET applications that reached a clinical stage for non-GB tumors are listed in Appendix A.

### 3.2. Vascular Endothelial Growth Factor Receptor (VEGFR)

#### 3.2.1. Current Status of VEGFR Targeted Therapy in GB

In GB, angiogenesis is primarily mediated by VEGF and generates blood vessels with distinctive features. GB tissues have shown to have very high levels of VEGF expression, associated with an up-regulation of the VEGFR2 [1,61,177]. This has led to clinical trials with either anti-VEGF antibodies (e.g., bevacizumab), VEGF binding proteins (e.g., aflibercept) or VEGF RTKIs (e.g., cediranib (AZD2171), vandetanib, pazopanib, vatalanib, sorafenib and tivozanib). Therapeutic exploitation of the VEGF axis has achieved substantial clinical benefit across many cancer subtypes, but results in GB remain disappointing [178]. *Bevacizumab* was granted accelerated approval by the FDA in 2009 for the treatment of patients with progressive or recurrent GB and remains the most extensively characterized anti-angiogenic GB treatment. Despite its approval, bevacizumab and numerous drug combinations have shown mixed results (see Table 4) [179,180,181]. Lomustine plus bevacizumab for progressive GB did reach phase III (EORTC 26101), but did not confer a survival advantage over treatment with lomustine alone [182]. The recombinant fusion protein *aflibercept* and the pan-VEGFR inhibitor *tivozanib* also appeared to have little activity in recurrent GB [183,184,185]. *Vatalanib* had initial good phase I results in GB patients, but a planned randomized phase II trial was discontinued [186,187,188]. *Cediranib*, a VEGFR-2 TKI, induced structural and functional normalization of tumor vessels and improved tumor blood perfusion for 1 month, which was associated with longer survival in GB patients [189]. However, cediranib used as a monotherapy or in combination with lomustine for recurrent GB in a phase III trial failed to improve progression-free survival (PFS) [190]. Phase II trials with the SM *axinitib* showed a comparable survival to patients treated with bevacizumab and the addition of lomustine or PD-L1 blocking avelumab did not result in improvement, despite the antitumor effect that was previously reported for the monotherapy in recurrent GB [191,192,193]. The inhibition of the VEGF receptor-mediated PKC activation by *enzastaurin* failed in randomized phase III trials in recurrent GB [194]. Multi-kinase inhibitors that also inhibit VEGF are later considered in Section 4 of this review.

In general, there is a need for validated markers to select patients who will likely benefit from anti-VEGF therapy. For example, the study of de Groot et al. indicates that circulating myeloid cells, such as VEGFR1+ monocytes, and myeloid-related cytokines are potential biomarkers for response to aflibercept in GB patients [195]. In addition, which drug combinations are optimal to be combined with a drug with anti-angiogenesis activity is unknown, but the pivotal role of the vasculature and angiogenic factors in the immunosuppression of GB supports the use of anti-angiogenics in combination with immunotherapy [196]. Alternative strategies include anti-integrin based approaches (e.g., cilengitide) which exceeds the scope of this review but have been reviewed elsewhere [197].

**Table 4 pharmaceuticals-14-00626-t004:** Clinical trials in GB targeting the vascular endothelial growth factor receptor.

Compound	Type	Clinical Trials: Phase, Overall Conclusion (+) or (−), (Combined or Compared Therapy)	Reference
Bevacizumab	mAb	II (+) (single/irinotecan)	[198]
		II (+) (single)	[199]
		II (+) (single)	[200]
		II (+) (TMZ)	[201]
		II (+) (TMZ)	[202]
		II (+) (TMZ)	[203]
		II (−) (TMZ)	[204]
		II (−) (RT/hypoRT)	[205]
		III (−) (RT/TMZ)	[181]
		III (−) (RT/TMZ)	[180]
		II (−) (RT/TMZ)	[206]
		II (+) (RT/TMZ)	[207]
		II (+) (RT/TMZ)	[208]
		II (−) (hypoRT/TMZ)	[209]
		II (−) (hypoRT/TMZ)	[210]
		II (+) (irinotecan)	[211]
		II (+) (irinotecan)	[212]
		II (−) (irinotecan/TMZ)	[213]
		II (+) (irinotecan/TMZ)	[214]
		II (−) (irinotecan)	[215]
		II (−) (irinotecan/TMZ)	[216]
		II (−) (irinotecan/TMZ)	[217]
		II (−) (cetuximab/irinotecan)	[100]
		II (−) (TMZ/lomustine)	[218]
		II (+) (lomustine)	[219]
		II (−) (lomustine)	[220]
		III (−) (lomustine)	[182]
		II (−) (carboplatin)	[221]
		II (−) (carboplatin/irinotecan)	[222]
		II (+) (rindopepimut)	[116]
		I/II (−) (BKM120)	[223]
		I/II (−) (dasatinib)	[224]
		II (+) (ERC1671 vaccine)	[225]
		II (−) (onartuzumab)	[226]
		II (−) (temsirolimus)	[227]
		II (−) (tandutinib)	[228]
		II (+) (fotemustine)	[229]
		II (−) (fotemustine)	[230]
		II (RT/TMZ/everolimus)	[231]
		II (−) (metronomic etoposide/TMZ)	[232]
		II (−) (panobinostat)	[233]
		I (+) (DEHSRT^#^)	[234]
		II (−) (sorafenib)	[235]
		II (−) (erlotinib/RT/TMZ)	[96]
		II (erlotinib)	[236]
		II (−) (vorinostat)	[237]
		I/II (−) (vorinostat/TMZ)	[238]
		II (−) (enzastaurin)	[239]
Cediranib (AZD-2171)	SM	II (+) (single)	[189]
		III (−) (lomustine)	[190]
		II (+) (gefinitib)	[90]
		II (active, not recruiting) (olaparib)	NCT02974621 [67]
Aflibercept	Fusion protein *	II (−)	[184]
		I (+) (RT/TMZ)	[185]
Vatalinib (PTK787/ZK222584)	SM	I (+) (imatinib/hydroxyurea)	[188]
		I (+) II (term) (RT/TMZ)	[187]
		I (+) (RT/TMZ/anti−epileptic drug)	[186]
Axitinib	SM	II (+)	[192]
		II (+) (lomustine)	[193]
		II (−) (avelumab)	[191]
Tivozanib	SM	II (−)	[183]
Ramucirumab	mAb	II (completed, no results) (IMC−3G3)	NCT00895180 [67]
Sorafenib	SM	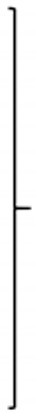	See Table 9	
Sunitinib	SM	
Nintedanib (BIBF 1120)	SM	
Pazopanib (GW786034)	SM	
Vandetanib (Caprelsa, ZD6474)	SM	
Cabozantinib (XL-184)	SM	
Regorafenib (BAY73-4506)	SM	
Dovitinib (TKI258)	SM	
Ponatinib (AP24534)	SM	
Lenvatinib (E7080)	SM	
Everolimus (AEE788)	SM	
Anlotinib (AL3818)	SM	

(*) recombinant fusion protein consisting of the extracellular domains of VEGFR1 and VEGFR2 fused to an immunoglobulin Fc domain, (^#^) Dose escalation study of hypofractionated stereotactic radiation therapy.

#### 3.2.2. VEGFR Radiopharmaceuticals

Molecular imaging could be a powerful tool for estimating the VEGF content within tumor tissues and select patients to benefit from VEGF targeted therapy. In addition, due to the issue of a ‘pseudo-response’ on contrast-enhanced magnetic resonance imaging (MRI) upon anti-angiogenic treatment of high-grade glioma, there is extensive interest in the development of PET and SPECT ligands that target specific aspects of the angiogenic process [240,241]. Clinically, [^89^Zr]Zr-bevacizumab has been used to visualize multiple malignancies, including breast cancer, renal tumor lesions and diffuse intrinsic pontine glioma [242,243,244]. The poor results of multiple clinical studies in GB patients with bevacizumab debate its rationale as a promising therapeutic radiopharmaceutical, but its role as an imaging biomarker is still underexplored. However, there was no significant uptake of [^89^Zr]Zr-bevacizumab in multiple intracranial tumor models, including a GB model [244].

In patients with histologically verified brain tumors, [^123^I]iodo-VEGF scintigraphy has shown to be promising for the visualization of tumor angiogenesis and may provide relevant prognostic information in patients with glioma [245]. Preclinically, [^111^In]In-/[^125^/^131^I]iodo-bevacizumab, [^89^Zr]Zr-ranibizumab and [^64^Cu]Cu-ramucirumab showed potential for different cancer types as listed in Appendix A [246,247,248,249,250]. However, due to the poor PK characteristics of mAbs, efforts have been put towards radiopharmaceuticals based on VEGF isoforms ([^64^Cu]Cu-VEGF(121/165)), linked to human transferrin ([^111^In]-hnTf-VEGF) or affibodies ([^111^In]-ZVEGFR2-Bp2), and peptide-based ([^64^Cu]Cu-VEGF(125/136) radiopharmaceuticals for imaging VEGFR expression in GB [251,252,253,254,255].

### 3.3. Mesenchymal-Epithelial Transition Factor (MET) Receptor

#### 3.3.1. Current Status of MET Targeted Therapy in GB

The hepatocyte growth factor (HGF) is the only known ligand for the MET receptor, which plays a major role in the progression, therapeutic resistance and recurrence of GB. It is commonly expressed in GB and strong MET expression was found in tumor cells, blood vessels and peri-necrotic areas of glioma samples [74,124,256]. High MET intensity correlates with high WHO grade and shorter PFS and OS [256,257]. Other reviews have provided comprehensive information on HGF/MET functions and modes of action in GB [256,258,259,260]. An overview of clinical trials in GB patients targeting MET is given in Table 5.

In recurrent GB, outcome to therapy with the monovalent MET inhibitor *onartuzumab* or the antibody against HGF rilotumumab (AMG102) combined with bevacizumab were discouraging [226,261,262]. Amgen also halted all clinical trials of rilotumumab after alarming effects occurred during the treatment of advanced gastric cancer [263]. The selective MET inhibitor capmatinib (INC280, Tabrecta™) has recently received approval by the FDA for treatment of metastatic MET-mutated NSCLC [264]. However, no clear clinical activity of capmatinib plus buparlisib was observed in adults with MET-amplified recurrent GB [265]. In particular, PLB-1001 raised expectation for success (following a positive phase I study), achieving responses in MET-altered glioma patients and supporting its further clinical investigation. PLB-1001, also called bozitinib, is an orally bioavailable ATP-competitive inhibitor of the proto-oncogene c-met and has shown to be highly selective and BBB-permeable [266]. Other ongoing clinical trials with MET inhibitors include a phase I trial on volitinib for recurrent or non-responding primary CNS tumors (NCT03598244) and a phase Ib trial on crizotinib in addition to the standard treatment for newly diagnosed GB (NCT02270034) [67]. However, crizotinib treatment in combination with dasatinib in children with high-grade glioma was poorly tolerated and its activity was minimal [267]. Potent c-MET kinase inhibitor SGX-523 and mAb against HGF, YYB-101, inhibited GB cell growth in vitro and in vivo. However, clinical trials registered for these agents to treat solid tumors have been terminated or no results could be found (NCT00606879, NCT00607399, NCT02499224) [67,256,268,269,270]. Finally, tivantinib, CM-118, INCB28060, altiratinib and foretinib were found effective against GB in vivo but clinical trials have not been initiated yet [256,268,271,272,273,274,275,276]. Multi-kinase inhibitors that also inhibit MET are covered in Section 4 of this review.

Based on these dissatisfying results, resistance of GB to single modality anti-MET drugs has been investigated and critical proteins that were altered in MET inhibitor-resistant GB include mTOR, FGFR1, EGFR, STAT3 and COX-2. In addition, targeted therapies against EGFR and VEGF in GB (including bevacizumab) often result in resistance due to activation of the MET signaling pathway. Simultaneous inhibition of MET and one of these upregulated proteins led to increased anti-GB effects both in vitro and in vivo [259,277,278,279,280].

**Table 5 pharmaceuticals-14-00626-t005:** Clinical trials in GB targeting MET (hepatocyte growth factor receptor).

Compound	Type	Clinical Trials: Phase, Overall Conclusion (+) or (−), (Combined Therapy)	Reference
Onartuzumab	mAb	II (-) (bevacizumab)	[226]
Rilotumumab (AMG102)	mAb	II (−)	[261]
		II (−) (bevacizumab)	[262]
Capmatinib (INC280)	SM	Ib/II (−) (buparlisib)	[265]
		I (active, not recruiting) (bevacizumab)	NCT02386826 [67]
PLB-1001 (bozitinib)	SM	I (+)	[266]
Volitinib (savolitinib)	SM	I (recruiting)	NCT03598244 [67]
Crizotinib (PF-02341066)	SM	I (active, not recruiting) (RT/TMZ)	NCT02270034 [67]
Cabozantinib * (XL184)	SM	See Table 9	

(*) also targets VEGFR2 + AXL.

#### 3.3.2. MET Radiopharmaceuticals

In recent years, many studies have shown that the expression level and activation status of MET are closely correlated to MET-targeted therapy response and clinical prognosis, thus highlighting the importance of evaluating the MET status during and prior to targeted therapy [27,281]. Due to a limited number of validated MET mAbs that work in formalin-fixed and paraffin embedded biopsy samples, immunohistochemical evaluation of MET expression is a challenge [282]. Since HGF has been recognized to have a high binding affinity and specificity to MET, initial studies on targeted molecular imaging of MET were therefore mainly based on HGF ligands [283]. The recombinant human HGF was [^64^Cu]-labeled ([^64^Cu]Cu-rh-HGF) and PET imaging revealed specific and prominent uptake of the radiopharmaceutical in MET positive U87MG GB tumors [284]. One concern for these radiopharmaceuticals based on the HGF ligand is their potential to stimulate tumor growth by activating c-met and competition with the endogenous ligand, hindering clinical translation [27,283,284]. Hence, the mAb-based PET radiopharmaceuticals [^89^Zr]Zr-onartuzumab and [^76^Br]Br-onartuzumab were developed that specifically target MET in vitro and in vivo. Using the U87MG GB model, [^89^Zr]Zr-onartuzumab achieved higher tumor accumulation (i.e., tumor/muscle ratios) and effectively visualized changes in MET expression [285,286]. Rilotumumab (AMG102) is an HGF-binding antibody that prevents its binding to MET and [^89^Zr]Zr-AMG102 was developed by Price and colleagues. The authors reported that this radiopharmaceutical selectively accumulated in tumors with high levels of HGF protein, including the U87MG GB tumor model in vivo [287]. Kim et al. introduced tumor imaging using a radiolabeled c-met-binding peptide ([^125^I]iodo-cMBP-GGG) in a glioma xenograft model—tumors were successfully visualized on SPECT also supported by using an amino acid linker that reduces hydrophilicity [288]. However, relatively unfavorable in vivo kinetics led to a subsequent design by the same group proposing a radiolabeled dual peptide ligand that would recognize both c-met receptor and the α(v)β(3)integrin; [^125^I]iodo-c(MBP)-c(RGDyK). Although image contrast and overall quality were improved in U87MG tumor xenografts compared to [^125^I]I-cMBP-GGG, further optimizations are needed [289].

Other MET specific compounds were radiolabeled and tested preclinically—so far only in other cancer types, listed in Appendix A. [^125^I]iodo-MET4 might be interesting to study for glioma since in a cohort of gliomas, MET4 reacted with 63% [282]. To our best knowledge, [^18^F]F-AH113804, a peptide-based c-met specific PET imaging agent, is the only radiopharmaceutical translated into humans for detection of locoregional recurrence of breast cancer [290].

### 3.4. Platelet-Derived Growth Factor Receptor (PDGFR)

#### 3.4.1. Current Status of PDGFR Targeted Therapy in GB

PDGF is a growth factor family of ligands and receptors known to activate PI3K, mitogen-activated protein kinase, Jak family kinase, Src family kinase and phospholipase C gamma signal transduction pathways. Several of these pathways have been causally linked to glioma formation. Human gliomas, especially GB, express all PDGF ligands and both the two cell surface receptors, PDGFR-α and -β, correlated with bad prognosis [291,292,293]. PDGFR-α amplification is found in nearly 15% of GBs and nearly half of these tumors also contain amplifications and/or mutations in EGFR or the MET gene [69,294]. Proneural GBs have shown to be enriched for activating mutations in the PDGFR-α gene and also comprise IDH mutated gliomas, whereas classical GB is enriched for EGFR amplification [65]. PDGFR-β has also been described to be the preferentially expressed type of PDGFR in glioma stem-cells (GSCs) [295].

Table 6 gives an overview of clinical trials in GB targeting PDGFR. Phase II trials on anti-PDGFR-α mAb olaratumab (IMC-3G3) in treating patients with recurrent GB were completed but no results were published (NCT00895180) [67]. A phase II study of tandutinib (MLN518), an inhibitor of PDGFR-β, was closed at interim analysis due to lack of efficacy in patients with recurrent GB. Combined with bevacizumab, therapy was as effective but more toxic than bevacizumab monotherapy [228,296]. Another PDGFR inhibitor, nilotinib (AMN107), was studied in a phase II study in PDGFR amplified malignant glioma, but preliminary results were discouraging [67,297]. Numerous multi-kinase inhibitors that include PDGFR as a target have been studied in GB, such as imatinib, dasatinib, regorafenib, sorafenib, sunitinib, ponatinib, nintedanib and lenvatinib. Unfortunately, only a few had beneficial activity for GB patients; more details are provided in Section 4 and Table 9. PDGFR inhibitors in a preclinical stage for GB include AG1433/AG1296 (tyrphostins), CP-673451 and CT52923, of which some have shown to retard cell growth and radiosensitize GB [298,299,300,301,302].

#### 3.4.2. PDGFR Radiopharmaceuticals

The in vivo visualization of PDGFRβ expression might help to select PDGFRβ targeting treatment and radiolabeled PDGFR TKIs could be used to assess their regional distribution and kinetics. As an example, [^11^C]C-Imatinib has shown potential to assess the distribution of imatinib in a study on baboons [303]. A PDGFRβ-binding affibody ([^111^In- or ^68^Ga]-radiolabeled Z09591) clearly visualized U87MG xenografts by way of PET-/ SPECT-imaging [304,305]. For other preclinical studies with PDGFR imaging agents for non-GB tumors, see Appendix A. These include the mAb [^64^Cu]Cu-D13C6, the affibody conjugate [^89^Zr]Zr-ZPDGFRβ, the radioiodinated 1-[303]piperidin-4-amine (IQP) and radiobrominated PDGFRβ ligands [306,307,308]. Recently, radiogallium-labeled peptides for PDGFRβ were developed and the effects of several linkers were studied, concluding that further probe modification is required [309]. Current radiolabeled multi-kinase inhibitors, targeting PDGFR amongst others, include [^99m^Tc-/^18^F-/^124^I]-labeled imatinib derivatives, imatinib mesylate ([^18^F]F-STI-571) and dasatinib derivatives ([^18^F]F-SKI249380, [^18^F]F-SU11248, [^11^C]C-sorafenib, [^18^F]F-sunitinib and [^11^C]C-nintedanib) [310,311,312,313,314,315]. The concept of [^18^F]F-STI-571 and [^18^F]F-SKI249380 containing micelles showed promising results in glioma-bearing animal models (see Section 4.3) [311,312].

### 3.5. Fibroblast Growth Factor Receptor (FGFR)

#### 3.5.1. Current Status of FGFR Targeted Therapy in GB

The evidence for the biological functions of fibroblast growth factor receptors (FGFR) in GB, as well as pharmacological approaches to target these receptors, have been recently reviewed [316]. A profound heterogeneity of FGFR1–4 expression across GB patients has been revealed by gene expression analysis. The strongest evidence by far indicates that FGFR1 has malignancy-promoting effects and FGFR1 signaling is linked to cancer stemness, invasion and radioresistance in GB [317,318,319]. Based on the evidence that FGFR1 is a GSC regulator, one can speculate if the inhibition of FGFR1 in combination with conventional therapy could prevent or delay GB recurrence [316]. However, the exact role of FGFR inhibitors in treatment of brain tumors remains elusive and the relevance of FGFR as a potential target is probably limited to 3% of GBs exhibiting fusions between FGFR and TACC (transforming acidic coiled-coil containing proteins) genes [316,320].

SMIs, such as lenvatinib, ponatinib, dovitinib, regorafinib, anlotinib and nintedanib, target multiple RTKs including FGFR (see Section 4 and Table 9). More selective FGFR inhibitors include erdafinitb (JNJ-42756493), AZ4547, PD173074, infigratinib (BGJ398) and futibatinib (TAS-120) (Table 7). Recent phase I trials evaluating erdafitinib, a highly selective pan-FGFR TKI, in advanced or refractory solid tumors showed tolerability and preliminary clinical activity, including in GB patients [321,322,323]. Multiple phase I/II/III trials on the FGFR1-4 irreversible inhibitor futibatinib are running in patients with advanced cancers harboring FGFR aberrations, with encouraging preliminary results in FGFR-1 mutant primary brain tumors [67,323]. BGJ398, a FGFR1-3 kinase inhibitor has been evaluated in a phase I trial in patients with advanced solid tumors harboring genetic alterations in FGFR [324]. In malignant glioma patients with FGFR–TACC fusion and/or activating mutation in FGFR1/2/3, two phase I/II trials with BGJ398 or AZD4547 were completed, but no results have been published [67]. Given the possibility that FGFR1-4 may have divergent functions in GB and considering the prevalent expression of these receptors on other cell types in the brain, the development of inhibitors with selectivity for individual FGFRs could be desirable. However, this is a challenge due to the high degree of homology of the kinase domains of FGFR1-4 [317].

#### 3.5.2. FGFR Radiopharmaceuticals

There is growing evidence that only a small fraction of cancer types may respond to FGFR inhibitors, emphasizing the need for non-invasive determination of the FGFR tumor status [326]. To the best of our knowledge, no FGFR-targeting radiopharmaceuticals have been studied for GB. A preclinical study should be noted using a *[^125^I]I-bFGF mAb*, which was capable of inhibiting the growth of hepatocellular carcinoma xenografts [327]. Interestingly, FGFR has shown to induce a radiosensitizing effect, a beneficial characteristic for EBRT and TRT [328].

### 3.6. Ehrin Receptors

#### 3.6.1. Current Status of Eph Receptor Targeted Therapy in GB

The Eph receptors are the largest family of RTKs, with 14 receptors divided into EphA and EphB subcategories, which bind to different types of ephrin ligands [329]. Eph/ephrin expression, clinical outcome and function in GB has been reviewed, including the use of Eph receptors as therapeutic targets [62,330,331]. A gradient of EphA receptor expression was observed in GB on the more de-differentiated stem-like cells and was absent on the less-aggressive differentiated tumor tissue [62]. The EphA2 and EphA3 receptors are highly expressed in GB cells and were found to promote the self-renewal infiltrative invasion of GSCs [330,332,333,334,335]. More recently, EphA3 was shown to be significantly elevated in recurrent versus primary GB [336]. The EphA4 and EphA7 receptors promote GB cell proliferation and migration by potentiating FGF receptor oncogenic signaling or promoting GSCs, respectively [62,330]. Contrarily, the function of EphB receptors is less well-characterized in GB but they clearly have functional roles in cell migration, invasion and tumor angiogenesis. EphB2 has emerged as a candidate for therapeutic strategies to prevent GB tumor invasion. However, inhibition of EphB2 signaling may also increase GB cell proliferation, as shown in preclinical studies [337,338].

Clinical trials in GB patients targeting Eph receptors are listed in Table 8. The anti-EphA3 mAB ifabotuzumab (KB004) was well tolerated and clinically active in a phase I study treating hematological malignancies and is currently being studied in a phase I trial in recurrent GB patients (NCT03374943) [67,339]. Preliminary findings reported a stable disease for 23 weeks in one of the cohorts [340]. A phase I/II trial evaluating a dendritic cell vaccine that includes an EphA2-binding peptide demonstrated measurable CD8+ T cell and clinical responses in 58% of recurrent glioma patients [341]. A phase I study on DS-8895a, an anti-EphA2 IgG1 mAb, was successfully completed in patients with advanced solid tumors but was not studied in GB yet (NCT02004717) [67,342]. Clinical results of multi-targeted TKIs that also inhibit EphA2 (dasatinib) or EphB4 (tesevatinib) are covered in Section 4 and Table 9.

Quite a few proven preclinical successes of Eph inhibition in GB are published [334,335,337,343,344,345]. Importantly, co-targeting multiple Eph receptors might be necessary to achieve a therapeutic response [62,330,346,347].

**Table 8 pharmaceuticals-14-00626-t008:** Clinical trials in GB targeting the Ephrin receptors and the insulin-like growth factor 1 receptor.

Target	Compound	Type	Clinical Trials: Phase, Overall Conclusion (+) or (−), (Combined Therapy)	Reference
Ephrin receptors	Tesevatinib (KD019/ XL647)	SM	See Table 9	
	Ifabotuzumab (KB004)	mAb	I (recruiting) (prelim)	[340]
				NCT0337494 [67]
	Dasatinib (BMS-354825)	SM	See Table 9	
IGF1R	Cixutumumab (IMC-A12)	mAb	I (withdrawn) (temsirolimus)	NCT01182883 [67]
	IGF-1R/AS ODN *	as-odn	0/I (+)	[348]
			0/I (+)	[349]
	PPP/AXL1717	SM	I (+)	[350]
			I/II (unknown recruitment status)	NCT01721577 [67]

* antisense oligodeoxynucleotide.

#### 3.6.2. Ehrin Receptor Radiopharmaceuticals

Different approaches towards Eph-targeting radiopharmaceuticals were recently reviewed [351]. Eph receptors are often expressed on migrating tumor cells, especially at the edge where GB cells are actively invading into the brain parenchyma. This observation has led to the investigation of Eph mAbs as potential imaging agents, which might accurately delineate tumor borders and better define areas of active invasion, allowing more complete resection and better patient outcomes [352]. Next to mAbs, high affinity SM TKIs that bind to the intracellular ATP-binding pocket within the kinase domain of Eph receptors are the ideal basis for the development of [^11^C]- or [^18^F]-containing radiopharmaceuticals. The main challenge remains the high conservation of the ATP-binding pocket, resulting in a usually low selectivity of such compounds. However, this could be advantageous in case of TRT to target multiple pathways concomitantly [351].

EphA3 targeted [^64^Cu]Cu-IIIA4 PET/CT imaging revealed specific tumor uptake in an orthotopic GB model and IIIA4 conjugated to maytansine induced a potent GB anti-tumor response [336]. IIIA4-DOTA was also radiolabeled with lutetium-177 ([^177^Lu]Lu-IIIA4) and this RIT strategy was effective to target both subcutaneous and orthotopic GB tumor bearing animals with minimal toxicity [336,353]. Adding an α-particle-emitting bismuth-213 to IIIA4 enhanced the therapeutic effect in leukemic models but was not studied in GB yet [354]. In the above-mentioned phase I trial of the anti-EphA3 mAb ifabotuzumab (formerly known as KB004 and a humanized version of IIIA4) in recurrent GB patients, therapy and follow-up is guided using [^89^Zr]Zr-KB004 ([^89^Zr]Zr-ifabotuzumab) PET (NCT03374943) [67,339].

The EphA2 RTK is overexpressed in GB while it is expressed at low levels in normal neural tissues, representing an attractive imaging target for delineation of tumor infiltration. EphA2-4B3, a mAb specific to human EphA2, was [^64^Cu]-labeled through conjugation to the chelator NOTA and effectively delineated tumor boundaries in three different GB mouse models [355]. [^64^Cu]Cu-DOTA-1C1, another radiolabeled EphA2 mAb, showed specific uptake in the U87MG GB in vivo model [352]. The humanized anti-EphA2 antibody DS-8895a was radiolabeled with [^111^In], [^125^I] and [^89^Zr], but because of its superior imaging and tumor uptake characteristics in breast cancer xenografts, [^89^Zr]Zr-DS-8895 was chosen as a lead compound. The results were superior to those of [^64^Cu]Cu-DOTA-1C1, particularly at later time points, when maximal uptake in human tumors is anticipated and when [^89^Zr] half-life is better suited for human trials [356]. A safety and bioimaging trial of [^89^Zr]Zr-DS-8895a in patients with advanced EphA2-positive cancers was completed, but no results have been published or shared to date (NCT02252211) [67].

Imaging biomarkers for EphB4 were developed due to their overexpression in both tumor cells and angiogenic blood vessels in GB [357]. Benzodioxolyl-pyrimidine-based radiopharmaceuticals targeting EphB4 and indazolylpyrimidinyl derivatives as high-affinity EphB4 receptor ligands were developed, but with discouraging results [358,359]. In contrast, a peptide-based radiopharmaceutical [^64^Cu]Cu-TNYL-RAW, was useful for PET/CT imaging of EphB4 receptor expression, with specific uptake in both U251 and U87MG GB tumor bearing mice [357,360]. Other EphR targeting peptides, including SWL and SNEW, have been radiolabeled (Appendix A) [351,361]. Finally, the recently developed [^18^F]-labeled xanthine derivatives have potential for PET imaging of Eph receptors but lack studies in GB to date [362].

### 3.7. Insulin-Like Growth Factor 1 Receptor (IGF1R)

#### 3.7.1. Current Status of IGF1R Targeted Therapy in GB

Insulin-like growth factors (IGFs) promote tumorigenesis and treatment resistance [363,364]. It was confirmed that IGF1 and its receptor (IGF1R) are overexpressed in GBs and could be used as a prognostic factor to identify shorter survival and less favorable response to TMZ [64,365,366,367]. IGF1R signaling has also been found to correlate with resistance to therapies that target other kinases including the EGFR, HER2 and mTOR. In GB patients, resistance to anti-EGFR therapies has been linked to IGFR [368,369]. A number of different strategies were developed: anti-IGF1R antibodies, IGF1/2 neutralizing antibodies, SMIs and IGF ligand TRAPs [363,364,370]. For an overview of IGF as a target for malignant glioma, see Trojan J et al. [371].

Good preclinical results for GB therapy were found for the following IGF1R inhibitors: PQ401, GSK1838705A, picropodophyllin/AXL1717, BMS-536924, BMS-754807, NVP-AEW541 and mAb IMC-A12 (cixutumumab) [1,367,372,373,374,375]. BMS-754807 exhibited potent antiproliferative effects on GB cell lines and was more effective than OSI-906 (linsitinib), which could be explained by off-target effects exerted on other protein kinases independently of IGF-1R inhibition [376].

However, despite preclinical efficacy of experimental IGF1R TKIs, most clinical trials reported an insignificant cancer curative value (e.g., dalotuzumab, robatumumab, R1507, figitumumab) [363,377,378,379,380,381]. In a pilot study in malignant glioma, a vaccine consisting of tumor cells pre-treated with an IGFR1 antisense oligodeoxynucleotide was found to elicit positive clinical responses in 8/12 patients [348]. A subsequent pilot vaccine trial confirmed its potential [349]. The mAb cixutumumab (IMC-A12) demonstrated (single-agent) activity in GB models and a favorable safety and PK profile in patients with advanced, resistant, solid tumors [367,382,383]. A phase I trial of cixutumumab in combination with temsirolimus in pediatric patients with recurrent or refractory solid tumors, including glioma, was withdrawn (NCT01182883) [67]. The best results are obtained with the semisynthetic cyclolignan picropodophyllin (PPP), the active agent in *AXL1717*, which interferes with the auto-phosphorylation of IGF1R. This drug increases the radiosensitivity of glioma GSCs and causes dramatic tumor regression in intracerebral xenografts, indicating passage of PPP across the BBB [375,384]. Clinically, AXL1717 induced responses in 4/9 (44%) of patients with relapsed malignant astrocytomas. A new formulation of the drug will be used in further investigations in order to better define the optimal dose [350,364].

#### 3.7.2. IGF1R Radiopharmaceuticals

The current status of IGF-1R molecular imaging in cancer was previously reviewed [385]. Multiple SMs (including IGF-1) and mAbs have been radiolabeled with Indium-111 for SPECT imaging of IGF1R, see Appendix A [386,387,388]. Prabhakaran et al. screened many TKIs and selected BMS-754807, a SM dual inhibitor of IGF1R/IR in phase III clinical trials for a variety of human cancers, as one of the candidate ligands for PET imaging. In healthy rodents, the radioligand [^18^F]F-BMS-754807 exhibited negligible brain uptake [389]. Autoradiography showed 5.25-fold higher binding of [^18^F]F-BMS-754807 in surgically removed GB tissues in comparison to control brain tissues [390]. In addition, the radiopharmaceutical [^11^C]C-GSK1838705A did penetrate the BBB and showed retention in the brain in vivo. The radioligand exhibited high uptake in U87MG cells with >70% specific binding to IGF1R [391].

Preclinical studies on cixutumumab (IMC-A12) radiolabeled with [^111^In] and [^225^Ac] have shown theranostic potential in breast cancer [392,393]. Several affibody-based radiopharmaceuticals for IGF1R PET imaging were able to discriminate between high and low IGF-1R-expression tumors and have the potential for patient selection for IGF-1R-targeted therapy (Appendix A) [394,395,396,397]. In a preclinical study in GB, the [^64^Cu]-labeled NOTA-conjugated affibody ZIGF-1R:4:40 specifically targeted IGF-1R in vitro and in vivo [397].

## 4. Multi-Kinase Inhibition for GB Therapy

GBs are known to have a high level of heterogeneity and often contain a mixture of cells with an amplification and activation of different RTKs. Single RTK inhibition often leads to transient responses and further disease progression due to compensation mechanisms of other pathways. For example, PDGF/PDGFR and FGF/FGFR pathways provide potential escape mechanisms from anti-VEGF/VEGFR therapy [178]. GB resistance to EGFR and MET inhibition could be overcome via blockade of FGFR-SPRY2 bypass signaling [398]. In addition, activation of RTKs not only drives PI3K/Akt signaling activation, but also stimulates other signaling pathways including MAPK, NF-*κ*B and STAT3 [399]. It is noteworthy that PIK3CA and EGFRvIII mutations often lead to PI3K activation independent of RTKs, such as EGFR, HER2, PDGFR, VEGFR and c-met, suggesting that inhibition of RTKs alone is not sufficient to obstruct PI3K/Akt signaling [400]. Therefore, simultaneous targeting of multiple RTKs might be more effective to treat GB, but one of the main challenges for its implementation in clinical practice remains toxicity.

### 4.1. Current Status of Single Agent Multi-Kinase Inhibitors for GB Therapy

Different multi-kinase inhibitors have been studied in GB without convincing results, including vandetanib, cabozantinib (XL-184) and dasatinib—Table 9 gives an overview, including the relevant targets [224,401,402,403,404,405]. A phase II trial provided evidence of clinical activity of cabozantinib in patients with recurrent GB naive to antiangiogenic therapy, although the results did not meet the predefined statistical target for success [403]. Dasatinib was ineffective in phase I/II trials in recurrent GB, even in combination with bevacizumab or lomustine, which could be due to a proposed active efflux mechanism [224,402,404,406]; however, efficacy of dasatinib treatment of PDGFRα-driven high grade gliomas could be enhanced with everolimus [407].

Regorafenib (BAY 73–4506) treatment showed positive results preclinically and an encouraging overall survival benefit in a phase II trial for recurrent GB (REGOMA trial) and warrants further phase III studies [408,409,410]. Regorafenib was also evaluated in a phase I trial in combination with cetuximab in patients with advanced cancer, including one GB patient, with evidence of clinical benefit [411]. Little beneficial activity was seen with imatinib (Gleevec) for GB patients despite clear efficacy in preclinical studies [412,413,414]. A phase II study of imatinib in combination with hydroxyurea generated promising results in GB and grade III malignant gliomas, but the positive treatment outcome could not be attributed solely to imatinib [415,416]. Tyrphostin (AG-1296) has similar targets but is still at a preclinical stage [417]. In a phase I/II trial, lapatinib did not show significant activity in recurrent GB patients and combination with the standard treatment for newly-diagnosed GB (tolerable and safe), but further investigations are required to evaluate its efficacy [418,419,420]. The anti-tumor activity of the phase II trial using lapatinib in combination with pazopanib was insufficient [421]. Neratinib is in phase III development for metastatic breast cancer and in phase I/II development for advanced breast cancer and other solid tumors, including GB [422]. Neratinib is also included in the recruiting INdividualized Screening Trial of Innovative Glioblastoma Therapy (INSIGhT) trial [423]. Sorafenib was approved by the FDA for the treatment of advanced renal cell carcinoma and has an adequate BBB penetration, but was not effective in GB patients, even not in combined treatment with TMZ, RT and others [235,424,425,426,427,428,429]. The nonspecific RTKI sunitinib did not offer significant improvement in the outcome of several GB clinical trials [430,431,432,433,434]. Furthermore, clinical phase II trials of the multi-kinase inhibitors ponatinib, lenvatinib (E7080) and nintedanib in patients with bevacizumab-refractory GB were associated with minimal activity [435,436,437,438]. Moreover, in a dose-escalation phase I study in patients with recurrent GB, everolimus (AEE788) administration was associated with unacceptable toxicity and minimal activity [87]. The combination with RAD001 resulted in clinically significant thrombocytopenia in GB patients [85].

Anlotinib is a more novel multitarget RTKI that was positively evaluated in multiple cancer types. Two case reports were published in GB patients, with a suggestion that FGFR3-TACC3 fusion could be a novel indication for treatment with anlotinib [439,440]. Multiple clinical trials in recurrent or newly diagnosed GB patients are recruiting to test anlotinib as a monotherapy or in combination with TMZ or the standard Stupp regimen (NCT04004975, NCT04547855, NCT04157478, NCT04119674) [67]. In phase I studies in patients with advanced solid malignancies, tesevatinib (KD019/XL647) was well tolerated up to the maximum tolerated dose [441]. A phase II study of tesevatinib monotherapy in patients with recurrent GB was completed—results are awaited to emerge (NCT02844439); however, the efficacy of tesevatinib monotherapy was relatively modest in the intracranial GBM12 model, despite excellent brain penetration [67,442].

SMIs that act on multiple RTKs and are in a preclinical stage for GB include amuvatinib (MP470) (c-KIT and PDGFRα) [443], PD173074 (FGFR/VEGFR) [444] and CUDC-101 (HER2 and EGFR) [445,446]. Finally, GB treatment with multivalent ligand-based vector proteins, also called QUAD, warrant further (pre)clinical development. As an example, Sharma et al. [447] developed a QUAD that targets four GB-associated receptors at the same time (IL-13RA2, EphA2, EphA3, EphB2), which was evaluated in both in vitro and in vivo experiments.

**Table 9 pharmaceuticals-14-00626-t009:** Clinical trials on single agent multi-kinase inhibitors for GB therapy.

Target	Compound	Type	Clinical Trials: Phase, Overall Conclusion (+) or (−), (Combined Therapy)	Reference
EGFR + HER2	Lapatinib (Tykerb, GW572016)	SM	II (−)	[418]
			I (+)	[420]
			Pilot II (+) (RT/TMZ)	[419]
			I/II (−) (pazopanib)	[421]
			II (ongoing) (RT/TMZ)	NCT01591577 [67]
			I (ongoing) (pre-surgery	NCT02101905 [67]
VEGFR-2 + EGFR + RET	Vandetanib (Caprelsa, ZD6474)	SM	I (+) (RT/TMZ)	[448]
			I/II (−)	[405]
			I (+) (Sirolimus)	[449]
			II (−) (RT/TMZ)	[450]
			Pilot (ongoing) (sunitinib, erlotinib)	NCT02239952 [67]
EGFR + HER1, HER2 and HER4	Neratinib (Nerlynx™)	SM	II (TMZ) (recruiting)	[423]
c-MET, VEGFR-2, RET, KIT, FLT3, AXL and TEK	Cabozantinib (XL-184)	SM	I (+) (RT/TMZ)	[451]
			II (modest) (received prior antiangiogenic therapy)	[401]
			II (+/−) (naive to antiangiogenic therapy)	[403]
			II (recruiting)	NCT02885324 [67]
VEGFR1–3 + TIE2 + KIT/RET/ RAF1/BRAF genes + PDGFR + FGFR + colony stimulating factor 1 receptor	Regorafenib (BAY73-4506)	SM	II (+) (vs. lomustine)	[408]
			II (active, not recruiting)	NCT02926222 [67]
SRC + KIT + PDGFR + EPHA2 + BCR-ABL fusion	Dasatinib (BMS-354825)	SM	I/II (−) (CCNU)	[404]
			II (−)	[402]
			I (−) (bevacizumab)	[224]
			I (+) (erlotinib)	[406]
PDGFRα/β + Bcr-Abl + c-FMS + c-Kit	Imatinib (Gleevec)	SM	II (+/−) (hydroxyurea)	[415]
			I (+) (vatalinib/hydroxyurea)	[188]
			I/II (−) (single)	[414]
			II (−)	[412]
			II (−) (RT/CCNU)	[413]
			II (−)	[452]
VEGFR2/3 + Raf + PDGFR + c-KIT + Flt-3	Sorafenib	SM	II (−) (RT/TMZ)	[428]
			II (−) (TMZ)	[426]
			II (+) (TMZ)	[453]
			II (−) (erlotinib−EGFR)	[425]
			II (−) (bevacizumab)	[235]
			I (+) (RT/TMZ)	[454]
			I/II (−) (temsirolimus)	[424]
			I (−) (tipifarnib)	[429]
			I (−) (RT/TMZ)	[427]
			I/II (active NR) (everolimus)	NCT01434602 [67]
VEGFR1-2 + PDGFRβ + FGFR1-2-3	Dovitinib (TKI258)	SM	I (+)	[455]
			II (−) (no/prior bevacizumab)	[456]
PDGFR + VEGFR + FLT3 + RET	Sunitinib	SM	I (−) (irinotecan)	[432]
			II (−)	[433]
			II (−) (prior bevacizumab)	[430]
			II (−) (RT)	[431]
			II (−)	[434]
VEGFR1/2/3 + PDGFRα/β + c-Kit	Pazopanib (GW786034)	SM	I/II (−) (lapatinib)	[421]
			II (−) (single)	[457]
PDGFR + VEGFR + Src + FGFR	Ponatinib (AP24534)	SM	II (−) (prior bevacizumab)	[438]
PDGFR α/β + FGFR 1-3 + VEGFR 1-3	Nintedanib (BIBF 1120)	SM	II (−) (single)	[435]
			II (−) (prior bevacizumab)	[436]
FGFR1-4, PDGFRβ, VEGFR1-3, RET, and KIT	Lenvatinib (E7080)	SM	II (modest) (prior bevacizumab)	[437]
EGFR + VEGF	Everolimus (AEE788)	SM	IB/II (−) (RAD001)	[85]
			I (−)	[87]
VEGFR1/2/3 + FGFR1/2/3 + c-Kit + Ret	Anlotinib (AL3818)	SM	Case report (+)	[440]
			Case report (+)	[439]
			I/II (recruiting)	NCT04004975 [67]
			II (recruiting) (TMZ)	NCT04547855 [67]
			I/II (recruiting) (RT/TMZ)	NCT04157478 [67]
EGFR + VEGFR + EphB4	Tesevatinib (KD019/XL647)	SM	II (completed, no results)	NCT02844439 [67]

### 4.2. Current Status of Combined RTKI Therapy for GB

Combined treatment with different RTKIs is being explored in GB, but with unsatisfying results so far (see Table 3, Table 4, Table 5, Table 6, Table 7, Table 8 and Table 9). In addition, RTKIs with or without radiation and chemotherapy have been tested in GB patients, including erlotinib/bevacizumab, cediranib/gefitinib, cetuximab/bevacizumab and lapatinib/pazopanib [90,96,100,421]. Multi-targeted approaches for the RTK family also include co-inhibition of anaplastic lymphoma kinase (ALK) and MET in MGMT-unmethylated GB patients, as well as targeting the immunoglobulin-like domains gene family in addition to PDGFRβ and EGFR [458,459]. It must be recognized that in the majority of solid tumors, the deregulation of the TK axis is in itself not the driver, but it occurs secondary to another molecular event that influences the expression of the ligands and/or receptors [363]. Hence, it may be necessary to prescribe an RTKI treatment in combination with drugs that target upstream pathways of RTKs and/or components within the downstream intracellular signaling cascades. The combination of TKIs targeting RTK and mTOR has been tested in GB patients with vandetanib/sirolimus and erlotinib/gefitinib/sirolimus [77,449,460]. Although the phase II trial of erlotinib plus sirolimus was well tolerated, negligible efficacy was noted among unselected recurrent GB patients [77]. Importantly, the class of dual mTOR agents are more likely to be effective in combined treatment strategies [461,462].

Several reviews have addressed the limited clinical efficacy of IGF1R inhibitors, the need for predictive biomarkers and selection of targets for co-inhibition [463,464]. Data suggest that inhibition of IGF1R potentiates the anticancer activity of pharmacological interference with the RAS/RAF/MEK/ERK pathway and PI3K/Akt/mTOR pathway signaling [364]. An example of a promising approach is the co-inhibition of EGFR and IGF1R, which improves the effect of CD95-ligand-induced apoptosis in GB cell lines [465]. Effectively inhibiting multiple pathways that are directly or indirectly involved in tumor angiogenesis could also increase effectiveness compared to single VEGFR inhibition. Hence, RTKI should be combined with inhibitors of the MAPK, JAK-STAT and PI3K/Akt/mTOR pathway [1,399,466,467]. A phase I clinical trial to test the combined treatment of cixutumumab (IGFR1) and selumetinib (MEK 1/2 inhibitor) was well-tolerated with preliminary evidence of clinical benefit but did not include GB patients [468]. Preclinical results treating GB with BKM120 (PI3K), sapanisertib (mTORC1/2) or NVP-BEZ235 (PI3K/mTOR) combined with MEK inhibitors were promising and may also be more potent in NF1-deficient GB [469,470,471]. Moreover, the combination of the mTORC1/2 inhibitor RES529 in combination with anti-angiogenic drugs, such as bevacizumab or sunitinib, provided encouraging results in GB preclinical/murine models [472]. However, the phase Ib/II study of BKM120, given in addition to the MET inhibitor INC280 in patients with recurrent GB bearing PTEN loss or MET alterations, was terminated (NCT01870726) [67]. Finally, a combined expression of EphA2, EphA3, EphB2 and IL-13RA2 is observed in almost every GB patient, presenting in tumor-infiltrating cells, tumor-initiating cells or GSCs and neovasculature. Therefore, a cytotoxic agent that simultaneously targets these four receptors could be powerful in destroying the tumor ‘ecosystem’ of GB [473]. A CED-administered cocktail of ephrin-A1 and IL-13-based cytotoxins has been extended to phase I trials for glioma therapy in dogs and is currently achieving encouraging clinical effects [474].

To conclude, the optimal combination of TK pathway inhibitors for GB treatment still needs to be determined and will mostly be very patient-specific. However, it seems plausible that the inhibition of both upstream RTK and downstream intracellular signaling cascades that include components of the MAPK, JAK-STAT and PI3K/Akt/mTOR pathway are likely to work synergistically and prevent resistance or unwanted feedback loops.

### 4.3. Multi-Kinase Targeted Radiopharmaceuticals

Derivatives or isotopologues of imatinib [303,310,312,475,476], sorafenib [315,477], lapatinib [478,479], sunitinib [313], nintedanib [23], cabozantinib [480] and vandetanib [481,482] have been radiolabeled and further developed preclinically to various degrees, yet no translation into the clinical setting was achieved (Appendix A). Multi-kinase inhibitors are likely to be of limited use to elucidate the expression of one individual RTK. On the other hand, the fact that multiple kinases may be overexpressed in the same cell is a desirable aspect—the translation of these polypharmacological inhibitors into PET radiopharmaceuticals can be seen as valuable [28]. When these ligands are radiolabeled with a therapeutic radionuclide in particular, several signal transduction pathways can be targeted simultaneously, which may increase their overall presence within the tumor and therefore affect this endogenous treatment effectiveness (including the increased radiation dose within the tumor). However, it is important to take into consideration that multi-kinase inhibitors might have different affinities for the individual RTKs, so one RTK might be targeted more efficiently than others. Another possible disadvantage of broad spectrum TKIs is that normal tissues might be targeted as well, resulting in additional, unwanted toxicity [23]. For example, [^18^F]F-cabozantinib demonstrated a high non-specific binding in tumor and unfortunately in heart tissue as well [480]. [^18^F]F-dasatinib (SKI249380) is currently under investigation in a clinical trial for potential diagnostic imaging in a wide range of solid tumors, but GB was excluded (NCT01916135) [67]. In a PDGF-B driven model of high-grade gliomas, [^18^F]F-dasatinib-PET favorably imaged CNS tumors by employing a nanocarrier-encapsulated formulation platform [311]. [^18^F]F-imatinib (STI-571) showed promising results to measure c-KIT expression levels in the U87MG GB model and further studies with [^131^I]I-STI-571 are underway [312]. [^64^Cu]Cu-vandetanib uptake was also noted in the U87MG GB model with an optimal tumor-to-muscle (T/m) ratio at ~5 h post injection but remained high (T/m >30), even until 24 h post-injection. Such ratios are significantly superior to T/m that typically occur with [^11^C]- or [^18^F]-labeled SM TKIs (T/m ±3–5) due to the allowable scanning time frame permitted by the half-life of those radioisotopes (typically 60–90 min p.i.) [28]. [^11^C]C-lapatinib differentially accumulated in brain metastasis vs. normal brain tissue, enabling tumor visualization [478]. Sorafenib was initially labeled at the carbonyl position by Asakawa et al. [477]. However, [^11^C]C-sorafenib injection in ABCB1a/1b; ABCG2^−/−^ mice confirmed that sorafenib brain accumulation is limited by both transporters, notwithstanding its tumor targeting potential that has been shown in three human cancer xenografts (head and neck, renal and breast cancer), expressing RAF1 kinase [315,483].

## 5. Selection and Radiolabeling of New TKIs for TRT of GB

Apart from the factors discussed in Section 2 for selecting an appropriate radionuclide and corresponding targeting ligand, biochemical and pharmacological characteristics also play an important role in the suitability of a TKI as a radiopharmaceutical. When considering brain imaging and TRT with TKIs, multiple physico-chemical properties including molecular weight, lipophilicity (required for binding of the inhibitor to the ATP binding pocket of the TK), polar surface area and hydrogen bond donors become increasingly important due to the restrictive BBB [17,28]. An algorithm including six physicochemical parameters for assessing the BBB permeability of CNS drugs was devised by Pfizer and is known as the multiparameter optimization algorithm (CNS MPO). A score of ≥4 indicates suitable parameters [484]. Based on the extensive literature review on RTKIs studied in GB (Table 3, Table 4, Table 5, Table 6, Table 7, Table 8 and Table 9), a selection process was applied to identify TKIs that have not yet been radiolabeled but have the potential to become GB TRT suitable agents. Table 10 gives an overview of the applied selection criteria.

The authors of this review also observed during the selection process, although this falls out of the scope of this review, that radiolabeled compounds targeting the PI3K/Akt/mTOR pathway (both for imaging and TRT) are scarce with almost no reports on relevant studies in GB and no clinical trials [485,486,487]. New radiopharmaceuticals are needed to image PI3K, Akt and mTOR signaling as current evaluation is limited to immunohistochemistry of patient samples [488].

This selection process revealed four SM TKI compounds that could potentially be converted into novel radiopharmaceuticals: BGJ398 (infigratinib), regorafenib, lenvatinib and neratinib (Figure 3). Based on the exclusion criteria, none of the mAb TKIs were found to be suitable. The selected SM RTKIs contain a halogen in aryl position that could indicate a possible location for radioiodination with iodine-125 (Auger emitter) or iodine-131 (beta emitter), and have a potential site for attachment of a chelator for complexation of a therapeutic radiometal, as summarized in Table 2. Noteworthy, regorafenib, lenvatinib and neratinib are multitargeted kinase inhibitors and therapeutic radiolabeling may lead to a multi-pathway inhibition. Concerning possible toxicity issues with these proposed new TRT agents, multiple factors play a role and prediction is difficult. However, it should be kept in mind that the concentration of the TKI as targeted therapy will be markedly higher than the prospective dosage given of a radiolabeled TKI during nuclear imaging or TRT. Additional considerations on GB TRT toxicity have been reviewed recently [17].

Radioiodination of the TKIs follows the same chemistry as that for non-radioactive (‘cold’) iodine. Nucleophilic substitution reactions occur with radioiodine anions while radioiodide can easily be oxidized to an electrophilic form for electrophilic substitution reactions. These electrophilic reactions are more favored as radiolabeling can be achieved in numerous ways without the need to directly generate carrier iodine. Radioiodine can be sourced from sodium iodide ([*I]NaI) molecules and oxidation of radioiodide can be achieved with a variety of mild oxidants such as peracetic acid, chloramine-T or N-chlorosuccinimide. These reactions tend to work on deactivated aromatic compounds, either directly via radio-deprotonation or via various demetallation reactions, yielding products with high in vivo stability. Radio-demetallation reactions make use of electropositive radioiodine and organometallic compounds, such as organoboron, organomercury and organothallium as well as metal compounds from group IV (Si, Ge, Sn). These usually yield products with high radiochemical purity in a highly regioselective manner. Halogen exchange (iodine for iodine) reactions are regularly used for the incorporation of radioiodine into organic molecules, with inorganic salts (ammonium sulfate) or copper (II) salts often being added to catalyze the iodine exchange. When radiolabeling the SM TKI with iodine, it is important to consider the effect that the larger halogen will have on the altered molecule and its behavior in vivo as the TKI may not retain its original biological properties [489].

Attachment of chelators to biomolecules is generally carried out through a nucleophilic reaction of a bifunctional chelating agent with an available primary amine [39,40]. The structures of most SM TKIs do not have free amines (primary or secondary) available for functionalization since the amines are in positions where they are required for binding to the TK active site. Therefore, in order to incorporate a metal chelator into the SM, an adjusted synthetic route for the SM is required. A number of SM TKIs have N-alkyl or O-alkyl substituents attached to their core framework which are not crucially involved with receptor binding. These locations would therefore be most suited for conversion to an N- or O-linker that is functionalized with a nucleophilic group for further reaction with a bifunctional chelator as previously described [40]. However, there are a number of challenges for attachment of a chelator to a SM TKI and subsequent radiometal complexation. The first consideration would be that the N- or O-linker needs to be of a sufficient length to place the chelator far enough from the TK binding motifs in order to not interfere with TK binding. The second consideration would be that the increase in size and molecular weight of the inhibitor could affect pharmacological properties, such as lipophilicity, metabolism and biological half-life, target binding and, crucial for GB targeting, its BBB crossing [28]. The described radiolabeling options, as well as the possible effect on the structure-activity–relationship of the four selected SM TKIs were further investigated.

BGJ398 (infigratinib) is an orally bioavailable selective pan-FGFR kinase inhibitor developed by Novartis Pharma AG (Basel, Switzerland) and currently licensed to BridgeBio Pharma (Palo Alto, CA, USA). BGJ398 is a modified polyurea, where the hydrogens are replaced by a 2,6-dichloro-3,5-dimethoxyphenyl, a methyl group and a 6-{[4-(4-ethylpiperazin-1-yl)phenyl]amino}pyrimidin-4-yl group. The 2,6-dichloro-3,5-dimethoxyphenyl moiety lowers the deconjugation energy, whereas the modified pyrimidinyl aniline is able to modulate the pharmacological profile. Results from computational docking studies suggest that the FGFR1-3 specificity of BGJ398 for FGFR kinases is derived from the coordination of the dichloro-dimethoxy phenyl group to the ATP binding site of the kinase [490]. Therefore, while BGJ398 does contain halogens as required by the selection criteria, conversion of the chlorides to a larger radioiodine would most likely affect the specific binding of BGJ398 to the FGFR, making radioiodination unfeasible. However, radiolabeling via attachment of a chelator group for complexation of therapeutic radiometals such as [^177^Lu] and [^90^Y] could be possible utilizing the piperazinyl side chain. The N-ethyl of this functionality protrudes out of the binding site and into the solvent region, thereby allowing for replacement of the ethyl with an alkyl linker connected to a chelator. Some consideration for application of BGJ398 is that its PK profile indicates a pH-dependent solubility that may alter the blood solubility and limit bioavailability. Results of the phase II trial in GB (NCT01975701) are awaited but since this compound is also being investigated for the treatment of other cancer types, radiolabeled BGJ398 could have applications beyond GB, especially for tumors with FGFR genetic alterations, such as cholangiocarcinoma, where BGJ398 reached a phase III trial (NCT03773302) [67]. Genetic changes such as FGFR2 fusions or FGFR mutations confer sensitivity to BGJ398-mediated FGFR inhibition, and in GB, FGFR-TACC fusions may serve as biomarkers [316,320,491,492]. A further consideration for BGJ398 is its BBB penetration. According to the CNS MPO algorithm, BGJ398 had a score of only 2.89, indicating limited feasibility for BBB crossing [484]. Therefore, since BGJ398 is not great for penetration of the BBB, this would possibly limit its potential as a GB imaging biomarker and a therapeutically radiolabeled BGJ398 would most likely need to be administered intracranially.

Regorafenib (BAY73-4506, Stivarga, Bayer HealthCare Pharmaceuticals Inc., Leverkuzen, Germany), which targets both membrane receptors (VEGFR, PDGFR, c-KIT, RET) and intracellular kinases (Raf, BRAF), is an oral multi-kinase inhibitor developed by Bayer which targets angiogenic, stromal and oncogenic receptor tyrosine kinase. The molecule is a generally flat, bi-aryl urea which mimics the adenine group of ATP that binds to the highly conserved ATP-binding pocket to inhibit kinase function [493]. For therapeutic isotope radiolabeling, the options are limited to possible attachment of a chelator in the position of the methyl group since the halides within the structure are located in the hydrophobic and allosteric pockets, making them unsuitable for conversion to radioiodine [494]. The amide hydrogen is required for hydrogen bonding and therefore attachment of a chelator moiety should be with an alkyl chain to minimize the interference of the binding of the compound to the enzyme active site [495]. Considerations for radiopharmaceuticals based on this RTKI is its effective half-life of 28 h and its high plasma protein binding. The compound is FDA-approved for advanced colorectal cancer, advanced gastrointestinal stromal tumor and hepatocellular carcinoma, and phase II/III trials are running in GB (NCT03970447, NCT02926222, NCT04051606) [67,496]. The most common grade 3–4 adverse reactions with the drug are hand/foot skin reactions, diarrhea, hypertension and fatigue [497]. The BBB penetration of regorafenib with a CNS MPO score of 2.44 is unlikely and its structural analog on sorafenib did not penetrate the BBB (consisting of a fluorine in the middle ring) [484,498]. It was also shown that brain accumulation of regorafenib is restricted by ABCG2 and ABCB1, hence inhibition of these transporters may be of clinical relevance for GB applications [499].

Lenvatinib (E7080), developed by Bayer Pharmaceuticals AG (Leverkuzen, Germany), targets FGFR1-4, PDGFRβ, VEGFR1-3, RET and KIT. The N-group of the quinoline moiety in Lenvatinib forms a hydrogen bond with the hinge region of the FGFR-1 receptor, while the two ureido N–H groups interact with the carboxylate side chain of αC-E531, residue. The ureido oxygen is then able to form a hydrogen bond within the binding pocket. Furthermore, the *N*-cyclopropyl moiety binds into the adjacent allosteric region of VEGFR2 and with the DFG motif adopting an ‘in’ conformation [494]. Therapeutic radiolabeling of lenvatinib will likely be limited to radiometal isotopes since the location of the chloride in the molecule within the hydrophobic pocket makes conversion to a larger radioiodine unfeasible. An amide moiety attached to the quinazoline core provides a potential site for modification to include a chelator, which would allow radiolabeling via chelator complexation. This amide will position itself just outside the ATP binding area and is therefore not involved with any hydrogen bonding [494]. Important considerations for lenvatinib as a prospective TRT agent are its excellent BBB penetration, its metabolization in the liver and its biological half-life of 28 h [500]. The ideal therapeutic radioisotopes to match this half-life would be rhenium-188, holmium-166, samarium-153 and copper-67 (Table 2). Radiolabeled lenvatinib is also worth exploring for other applications, since this SM has been approved for differentiated thyroid cancer, hepatocellular carcinoma and renal cell carcinoma as a single agent or in combination treatments. In addition, lenvatinib has shown promise in several other tumor types including medullary, anaplastic thyroid, adenoid cystic and endometrial cancer [501,502,503].

Neratinib (Nerlynx^TM^, Puma Biotechnology, Inc., Los Angeles, CA, USA), targets the EGFR and HER1, HER2 and HER4 receptors [494]. Neratinib has a highly reactive Michael acceptor that forms a covalent bond with a cysteine residue located within the ATP binding pocket of the EGFR protein and is therefore an irreversible inhibitor [504]. This poses an advantage for therapy since a therapeutically radiolabeled neratinib would remain bound to its target thereby increasing its therapeutic effect. Neratinib can potentially be radiolabeled with radiometal isotopes by chelation, whereby one of the methyl groups on the tertiary amine can be replaced with an alkyl-linked chelator. This amine is found to penetrate the solvent region and can therefore sustain modification without affecting the TK binding significantly. The chloride cannot be converted to radioiodine since it is required in the hydrophobic pocket [505]. Neratinib is FDA-approved to treat HER2-positive metastatic breast cancer and is included in the ongoing biomarker-based INSIGhT trial for newly diagnosed unmethylated GB (NCT02977780) [423]. Neratinib showed adequate penetration of the BBB and efficacy against brain metastasis, with a biological half-life of 28 h [506,507]. Neratinib specifically binds the drug-binding cavity of ABCB1, thereby reducing drug efflux and enhancing drug sensitivity, particularly in the brain [508]. The main toxicity of neratinib in clinical trials is gastro-intestinal and is essentially limited to diarrhea [509].

In summary, the selection of the four compounds was based on the inclusion and exclusion criteria stated in Table 10. Further evaluation of these selected compounds for GB TRT has revealed certain limitations. Firstly, therapy via radioiodination would mostly be unfeasible because of the location of the structural halogens in a TK binding site. However, this does not preclude the possibility of radiofluorination and the evaluation of these selected inhibitors for GB imaging. Secondly, functionalization of the selected inhibitors with a chelating agent will increase the size of the compounds, which may affect lipophilicity and BBB penetration. However, even if BBB passage is unlikely, despite deficits in barrier integrity in GB, CED or similar loco-regional drug application could be applied for TRT applications. CED has been applied in the clinic and was proved to be a safe and effective drug delivery method for GB with limited systemic toxicity [510]. Most RIT strategies for malignant gliomas were administered locally into the postsurgical cavity or intratumorally and the first results on the local administration of [^213^Bi]Bi-substance P and [^225^Ac]Ac-substance P are promising [511,512,513,514]. Overall, GB management would greatly benefit from the introduction of TRT using TKI-based radiopharmaceuticals.

Finally, we would like to point out that besides these RTKs that contain intrinsic enzyme activity, there are numerous other plasma membrane receptors that can influence downstream intracellular signaling pathways which play a major role in GB persistence and progression. These include the folate receptors and the NK-1 receptor that lead to development of radiolabeled folate-chelate conjugates and radiolabeled substance-P [511,514,515]. The advantage of TKIs is that many are designed to be multi-targeted, affecting multiple signaling pathways concomitantly. However, every radiopharmaceutical has its pro and cons and the ideal strategy for clinical establishment is a work in progress, particularly for GB [17].

## 6. Conclusive Statements

This review presented and critically assessed the current status of RTKIs for the treatment, nuclear imaging and targeted radionuclide therapy of GB. Tyrosine kinase receptors with particular relevance in the treatment of GB have been investigated, namely EGFR, VEGFR, MET, PDGFR, Eph receptor and IGF1R. This overview revealed that only a limited number of developed RTKIs have been explored for their potential theranostic application to date. Hence, through application of relevant selection criteria, four small molecule RTKIs are proposed as new radiopharmaceuticals: infigratinib, regorafenib lenvatinib and neratinib (Figure 3). These new radiopharmaceuticals have the potential to improve TKI targeted therapy patient selection via molecular imaging and to result in a more complete pathway inhibition via TRT and prediction of treatment response.

## Figures and Tables

**Figure 1 pharmaceuticals-14-00626-f001:**
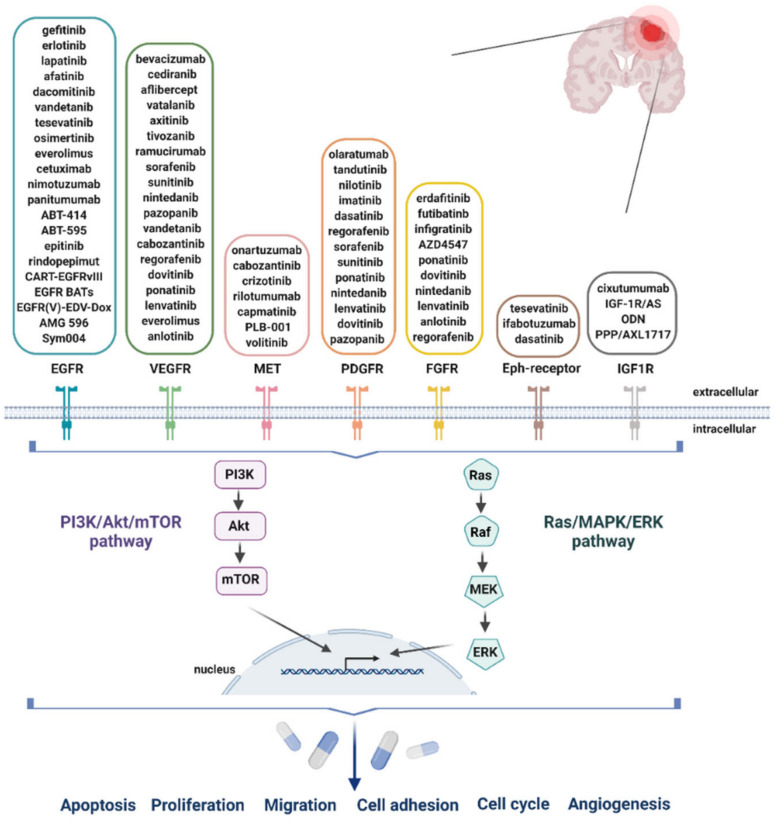
Overview of receptor tyrosine kinase inhibitors (RTKI) for the treatment of glioblastoma (GB).

**Figure 2 pharmaceuticals-14-00626-f002:**
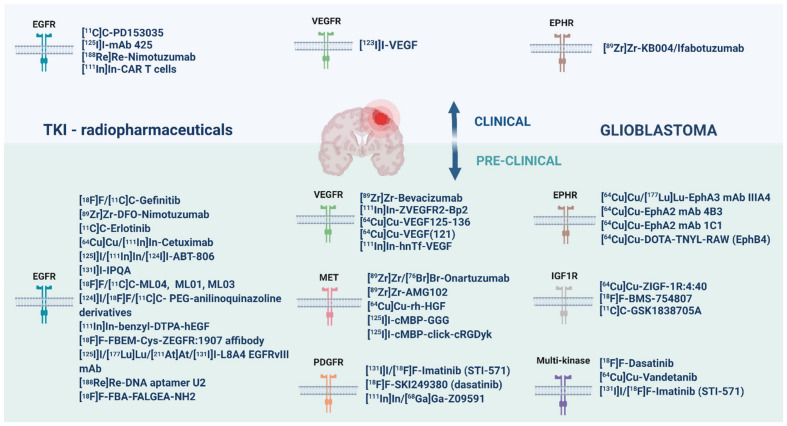
Overview of RTKI radiopharmaceuticals in GB. Info on organic radionuclides and radiometals commonly used for nuclear imaging and therapy purposes can be found in Table 1 and Table 2.

**Figure 3 pharmaceuticals-14-00626-f003:**
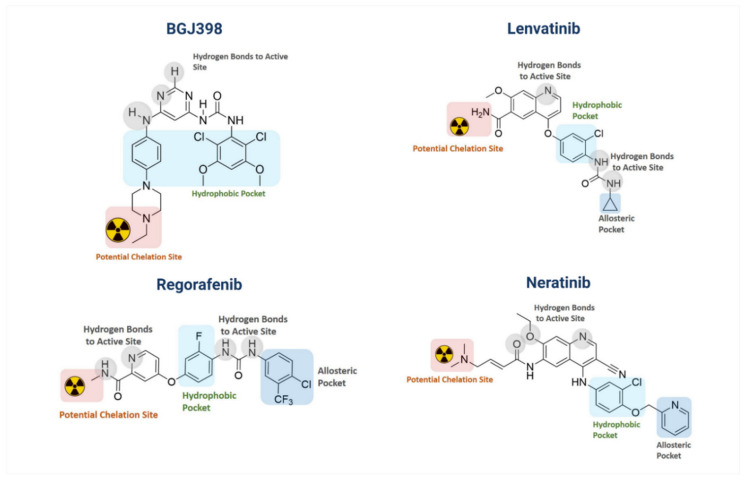
Chemical structures of the selected SM TKIs. The structure–activity relationship and potential radionuclide attachment sites are indicated.

**Table 1 pharmaceuticals-14-00626-t001:** Organic radionuclides commonly used for nuclear imaging and therapy purposes. Adapted from ref. [32].

Radionuclide	Half-Life	Energy (keV)	Function
^11^C	20.4 min	960 (β^+^)	PET
^13^N	9.96 min	1190 (β^+^)	PET
^15^O	2.07 min	1720 (β^+^)	PET
^18^F	119 min	640 (β^+^)	PET
^123^I	13.2 h	159 (γ)	SPECT
^125^I	60.1 h	15 (Auger)	Therapy
^131^I	8 d	365 (γ), 606 (β^−^)	SPECT and Therapy

**Table 2 pharmaceuticals-14-00626-t002:** Radiometals commonly used for nuclear imaging and therapy purposes. Adapted from ref. [30,33].

Radionuclide	Half-Life	Energy (keV)	Function
**Diagnostic**			
^64^Cu	12.7 h	656 (β^+^)	PET
^67^Ga	78.3 h	6.26 (Auger);	SPECT (Therapy)
		93, 184, 300, 393 (γ)	
^68^Ga	67.7 min	1899 (β^+^)	PET
^86^Y	14.7 h	1221 (β^+^)	PET
^89^Zr	78.4 h	902 (β^+^)	PET
^99m^Tc	6.02 h	140 (γ)	SPECT
^111^In	67.2 h	6.75 (Auger); 171, 245 (γ)	SPECT (Therapy)
^44^Sc	3.97	632 (β^+^)	PET
**Therapeutic**			
^67^Cu	2.58 d	141 (β^−^)	β-Therapy
		91, 93, 185 (γ)	
^89^Sr	52.7 d	1463 (β^−^)	β-Therapy
^90^Y	64 h	2280 (β^−^)	β-Therapy
^117m^Sn	13.6 d	150 (β^−^)	β-Therapy
^153^Sm	46.5 h	640; 710; 808 (β^−^)	β-Therapy
		103 (γ)	
^161^Tb	6.89 d	154 (β^−^)	β/AE-Therapy
		49, 75 (γ)	
		≤50 (AE)	
^166^Ho	26.8 h	665 (β^−^)	β-Therapy
		81 (γ)	
^169^Er	9.4 d	350 (β^−^)	β-Therapy
^177^Lu	6.75 d	176, 384, 497 (β^−^)	β-Therapy
		113; 208 (γ)	
^186^Re	3.7 d	1069 (β^−^)	β-Therapy
		137 (γ)	
^188^Re	17 h	2120 (β^−^)	β-Therapy
		155 (γ)	
^211^At	7.2 h	5870 (α)	α-Therapy
^212^Pb	10.2 h	570 (β^−^);	α-Therapy
		6050, 6090 (α—from ^212^Bi daughter)	
		238, 300 (γ)	
^213^Bi	45.6 min	5558, 5875 (α)	α-Therapy
		324 (γ)	
^223^Ra	11.4 d	5433 (α)	α-Therapy
		144, 154, 269, 324, 338 (γ)	
^225^Ac	10 d	5830, 5792, 5790, 5732 (α)	α-Therapy
		86, 440 (γ)	
^47^Sc	3.35 d	162 (β^−^)	β-Therapy

**Table 6 pharmaceuticals-14-00626-t006:** Clinical trials in GB targeting the platelet-derived growth factor receptor.

Compound	Type	Clinical Trials: Phase, Overall Conclusion (+) or (−), (Combined Therapy)	Reference
Olaratumab (IMC-3G3)	mAb	II (completed-no results) (ramucirumab)	NCT00895180 [67]
		II (+/−) (bevacizumab)	[228]
Tandutinib (MLN518)	SM	I/II (−)	[296]
Nilotinib (AMN107)	SM	II (completed, no results)	NCT01140568 [67,297]
Imatinib (Gleevec)	SM	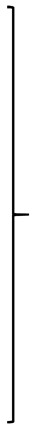	See Table 9	
Dasatinib (BMS-354825)	SM	
Regorafenib	SM	
Sorafenib	SM	
Sunitinib	SM	
Ponatinib	SM	
Nintedanib (BIBF 1120)	SM	
Lenvatinib (E7080)	SM	
Dovitinib (TKI258)	SM	
Pazopanib (GW786034)	SM	

**Table 7 pharmaceuticals-14-00626-t007:** Clinical trials in GB targeting the fibroblast growth factor receptor.

Compound	Type	Clinical Trials: Phase, Overall Conclusion (+) or (−), (Combined Therapy)	Reference
Erdafitinib (JNJ-42756493)	SM	I (+)	[321]
		I (+) (advanced or refractory solid tumors)	[325]
Futibatinib (TAS-120)	SM	I (+) (advanced solid tumors)	[323]
		I/II (active, not recruiting)	NCT02052778 [67]
Infigratinib (BGJ398)	SM	II (completed, no results)	NCT01975701 [67]
AZD4547	SM	I/II (completed, no results)	NCT02824133 [67]
Ponatinib (AP24534)	SM	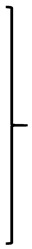	See Table 9	
Dovitinib (TKI258)	SM	
Nintedanib (BIBF 1120)	SM	
Lenvatinib (E7080)	SM	
Anlotinib (AL3818)	SM	
Regorafenib (BAY73-4506)	SM	

**Table 10 pharmaceuticals-14-00626-t010:** Selection criteria for potential novel TRT of GB.

**Inclusion Criteria**
TKI was studied in clinical trials for GB TKI is a mAb that can be conjugated to a chelator for use with metallic therapeutic isotopes, whereby the targeting/uptake will not be affected OR TKI is a SM that Contains a halogen which indicates a position that can potentially be radioiodinated Has a potential site for attachment of a chelator that will not drastically affect the structure–activity relationship of the inhibitor with the receptor binding site
**Exclusion Criteria**
TKI (SM and mAbs) has already been radiolabeled (diagnostic or therapeutic radionuclide) TKI SM does not contain a halogen or any possible site for chelator attachment Clinical trials results exclusion criteria: If results of clinical trials in GB patients reveal unwanted safety/tolerability issues, serious adverse events that were irreversible or responsible for treatment discontinuation If results of clinical trials in GB patients reveal unfavorable pharmacokinetic properties If results of clinical trials in GB patients did not result in a significant improved PFS and/or OS

## Data Availability

Data sharing not applicable.

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
