# Peer review of "Novel Receptor Tyrosine Kinase Pathway Inhibitors for Targeted Radionuclide Therapy of Glioblastoma"

_pharmaceuticals, 2021, doi:10.3390/ph14070626_

Round 1
Reviewer 1 Report
Review: Novel receptor tyrosine kinase pathway inhibitors for targeted radionuclide therapy of glioblastoma.
This paper reviews the present status of RTKIs used for treatment or imaging in glioblastoma. The review methodology is sound focusing on seven TKRs, status of the receptors in therapy and as radiopharmaceutical targets.
Below are two vey minor comments.
Line 26: Says seven TKRs but only lists six.
Line 167: Sentence starting with “For radiolabeling of….” Seems to be missing words to become a full sentence.
Author Response
Response to Reviewer 1 Comments
Point 1: Line 26: Says seven TKRs but only lists six.
Response 1: We would like to thank the reviewer for this particular comment. One of the RTKs was indeed missing in the list in the abstract. We added FGFR.
Point 2:Line 167: Sentence starting with “For radiolabeling of....” Seems to be missing words to become a full sentence.
Response 2: We agree with the reviewer that the sentence was not complete. We changed the sentence to:For radiolabeling of mAb fragments, the short-lived positron emitters, such as gallium-68 (t½ 1.13 h), copper-64 (t½ 12.7 h), yttrium-86 (t½ 14.7 h), and bromine-76 (t½ 16.2 h) are available to facilitate in vivo imaging.
Reviewer 2 Report
The reviewed publication is devoted to application of tyrosine kinase inhibitors for radionuclide diagnosis and therapy of glioblastoma. It is an important field of nuclear medicine, because glioblastomas are one of the most aggressive and difficult to treat cancers. The review is well prepared. I have no significant comments to it. However, I believe that in the conclusion the authors should add a comparison of the use of tyrosine receptors with the use of other approach, for example application of NK1 or folate receptors.
Small remarks:
Table 2. Sc-44 for imaging and Sc-47 should be added. This theranostic pair of radionuclides is becoming more and more popular.
In the case of Tb-161 Auger emission Auger electron emission should be included.
line 1097. The results of very promising studies on the use of 225Ac-substance P in the treatment of gliomas should be mentioned. e.g. L.Krolicki et all Semin.Nucl. Med. 2020,50(2):141-151.
Author Response
Response to Reviewer 2 Comments
Point 1: The reviewed publication is devoted to application of tyrosine kinase inhibitors for radionuclide diagnosis and therapy of glioblastoma. It is an important field of nuclear medicine, because glioblastomas are one of the most aggressive and difficult to treat cancers. The review is well prepared. I have no significant comments to it. However, I believe that in the conclusion the authors should add a comparison of the use of tyrosine receptors with the use of other approach, for example application of NK1 or folate receptors.
Response 1: We would like to thank the reviewer for this particular comment. We included a paragraph at the end of section 5 (Line 1101-1109):
Finally, we would like to point out that besides these RTKs that contain intrinsic enzyme activity, there are numerous other plasma membrane receptors that can influence downstream intracellular signaling pathways which play a major role in GB persistence and progression. These include the folate receptors and the NK-1 receptor that lead to development of radiolabeled folate-chelate conjugates and radiolabeled substance-P [511,514-515]. The advantage of TKIs is that numerous are designed to be multi-targeted, affecting multiple signaling pathways concomitantly. However, every radiopharmaceutical has its pro and cons and the ideal strategy for clinical establishment is a work in progress, particularly for GB [17].
Reference 515 was added to the manuscript.
515. Müller, C.; Schibli, R. Prospects in folate receptor-targeted radionuclide therapy. Front Oncol2013, 3, 249, doi: 10.3389/fonc.2013.00249.
Point 2: Table 2. Sc-44 for imaging and Sc-47 should be added. This theranostic pair of radionuclides is becoming more and more popular.
Response 2: We agree with the reviewer that this theranostic pair deserves a place in Table 2. 44Sc has indeed been investigated for PET imaging, because it decays by the emission of positrons (average energy, 632 keV), with a half-life of 3.97 h. 47Sc was proposed as a potential therapeutic match due to its low-energy β− emissions with decay characteristics (T1/2, 3.35 d; average energy, 162 keV) that are potentially useful for radionuclide tumor therapy. Hence this theranostic pair was added to the table.
Point 3: In the case of Tb-161 Auger emission Auger electron emission should be included.
Response 3: Thanks for this important remark. 161Tb indeed yields a significant number of short-ranging Auger/conversion electrons next to its (β-) and (γ) emissions. We added this to the table: (≤50 keV) (AE).
Point 4: Line 1097. The results of very promising studies on the use of 225Ac-substance P in the treatment of gliomas should be mentioned. e.g. L.Krolicki et all Semin.Nucl. Med. 2020,50(2):141-151.
Response 4: Line 1097: Reference number 511 by Królicki et al. in 2019 on the resultsof 213Bi-DOTA-substance P was included. This did not include the successful results of the same group on 225Ac-substance P. We agree with the reviewer and included 225Ac-substance P in the sentence with the corresponding reference (nr 514).